# A role for PVRL4-driven cell–cell interactions in tumorigenesis

**Natalya N Pavlova[1,2], Christian Pallasch[3†], Andrew EH Elia[1,2,4], Christian J Braun[3], Thomas F Westbrook[5], Michael Hemann[3], Stephen J Elledge[1,2]\***

[1]Department of Genetics, Harvard Medical School, Boston, United States; [2]Division of Genetics, Howard Hughes Medical Institute, Brigham and Women's Hospital, Boston, United States; [3]Department of Biology, Koch Institute for Integrative Cancer Research, Massachusetts Institute of Technology, Boston, United States; [4]Department of Radiation Oncology, Massachusetts General Hospital, Boston, United States; [5]Verna and Marrs McLean Department of Biochemistry and Molecular Biology, Department of Molecular and Human Genetics, Department of Pediatrics, Dan L. Duncan Cancer Center, Baylor College of Medicine, Boston, United States

**Abstract** During all stages of tumor progression, cancer cells are subjected to inappropriate extracellular matrix environments and must undergo adaptive changes in order to evade growth constraints associated with the loss of matrix attachment. A gain of function screen for genes that enable proliferation independently of matrix anchorage identified a cell adhesion molecule PVRL4 (poliovirus-receptor-like 4), also known as Nectin-4. PVRL4 promotes anchorage-independence by driving cell-to-cell attachment and matrix-independent integrin β4/SHP-2/c-Src activation. Solid tumors frequently have copy number gains of the PVRL4 locus and some have focal amplifications. We demonstrate that the transformation of breast cancer cells is dependent on PVRL4. Furthermore, growth of orthotopically implanted tumors in vivo is inhibited by blocking PVRL4-driven cell-to-cell attachment with monoclonal antibodies, demonstrating a novel strategy for targeted therapy of cancer.

*For correspondence: selledge@
genetics.med.harvard.edu

†Present address: Department I
of Internal Medicine, University
Hospital Cologne and Center for
Integrated Oncology Koln-Bonn,
Cologne, Germany

**Competing interests:** The
authors declare that no
competing interests exist.

**Reviewing editor**: Louis Staudt,
National Cancer Institute, United
States

## Introduction

As many as 90% of all human cancers originate from epithelial tissues. Epithelia have a distinct ability to form and maintain highly organized monolayers, which is reflected in their role in providing the inner lining of hollow organs. This unique architecture is dictated by the requirement for an epithelial cell to be physically anchored on a basement membrane, an organizing substratum composed of specific extracellular matrix (ECM) molecules. Cells physically attach to ECM via integrins, a class of signaling molecules that serve to stimulate the survival and proliferation of cells in a matrix attachment-dependent manner (*Hynes, 2002*). Conversely, loss of contact with the proper ECM molecules results in initiation of a cell death program known as anoikis (*Frisch and Screaton, 2001*), and other constraints on cellular expansion.

Early stages of epithelial cancer progression are universally characterized with genetic changes that confer ability to survive and proliferate in the absence of an appropriate matrix anchorage, which allows cellular expansion in a geometrically unconstrained manner. Though acquired early, the ability to tolerate the loss of anchorage remains critical for the survival of cancer cells throughout the course of disease progression, encompassing stages such as invasion of the underlying stroma, extravasation into blood vessels, survival in the bloodstream, and, eventually, metastatic outgrowth at a distant site with a distinct matrix composition.

Along with the loss of the requirement for anchorage, a propensity for self-aggregation is a characteristic of aggressive cancer cells. Thus, tumor-derived subclones with greater metastatic capacity in

**eLife digest** Epithelial tissue is one of the four major types of tissue found in animals, and is the only type of tissue that is able to form and maintain layers of cells that are just one cell thick. These layers provide inner linings to various cavities and hollow organs throughout the body—including the lungs and glandular organs such as mammary glands. A single-cell layer of epithelium is separated from the tissues beneath it by a supporting substance called the extracellular matrix. The individual cells within a single-cell layer are physically attached to the matrix, and when displaced from it, they promptly undergo programmed cell death. This mechanism preserves the single-cell layer pattern throughout the body and prevents epithelial cells from growing in inappropriate locations.

It is estimated that up to 90% of cancers in humans originate in epithelial tissue, and the cells within such tumors are known to survive and divide even when they are no longer attached to the extracellular matrix. Understanding how cancerous cells gain this ability may lead to new approaches to stopping tumor cells from dividing and colonizing tissues around the body.

To address this problem, Pavlova et al. explored which genes enable epithelial cells from the human mammary gland to grow without being attached to the extracellular matrix. They found that the gene that codes for a protein called poliovirus receptor-like 4 (PVRL4) allows attachment-free cell growth and also makes cells cluster together once detached from the matrix.

Normally, the *PVRL4* gene is not active in breast epithelial cells, but its activity is detected in many breast, lung, and ovarian tumors. Moreover, cancerous cells tend to cluster together when they are detached from the extracellular matrix. This behavior is particularly evident in the cells that divide aggressively to form tumors that subsequently migrate and colonize other tissues around the body. When Pavlova et al. used genetic techniques to silence PVRL4 in cells from breast tumors, they found that it reduced the formation of clusters by the cancer cells and also reduced their ability to grow in the absence of attachment.

Pavlova et al. also showed that interactions between the PVRL4 in one cell and a related protein called PVRL1 in a neighboring cell were responsible for holding the cells together in clusters. Moreover, PVRL4 triggers a form of signaling between the cells called integrin β4 signaling that allows them to survive without being anchored to the extracellular matrix.

Finally, Pavlova et al. found that injecting anti-PVRL4 antibodies (mouse proteins that attach to PVRL4 and prevent the formation of clusters) slows down the growth of breast tumors in mice. These findings suggest that inhibiting PVRL4 action with antibodies can be used as a new approach to the treatment of breast, lung, and ovarian cancers in humans.

vivo display increased self-aggregation in vitro; at the same time, subclones selected for increased in vitro aggregation were found to be more metastatic in mice (*Updyke and Nicolson, 1986*; *Saiki et al., 1991*). Invasion of the underlying stroma is frequently undertaken by large groups of tumor cells, a phenomenon known as collective, or cohort, cell migration (*Friedl and Gilmour, 2009*). Clusters of circulating tumor cells (CTCs) have been identified from the blood samples of breast, colorectal, prostate, and lung cancer patients as well as from mouse tumor models (*Molnar et al., 2001*; *Stott et al., 2010*; *Hou et al., 2011*). In particular, one report demonstrated that, on average, 50% of all breast and lung CTCs exist in circulation as aggregates (*Cho et al., 2012*). An increase in clustering behavior is not limited to self-aggregation, as interactions with a variety of other cell types have been shown to be essential for the dissemination of cancer cells and subsequent metastatic colonization. Thus, heterotypic interactions of cancer cells with the endothelial lining within the target organ microvasculature were documented as an initiating event for metastatic lesion formation (*Al-Mehdi et al., 2000*). Moreover, signaling elicited by the physical association of cancer cells with other cell types, such as platelets and macrophages, has been demonstrated to be essential for successful seeding and metastatic outgrowth (*Chen et al., 2011*; *Labelle et al., 2011*). Taken together, tumor-specific cell–cell contacts can provide a multifaceted survival advantage throughout the course of pathologic progression. Targeted therapies directed towards blocking such cell–cell contacts may therefore represent a novel cancer treatment approach.

In this study, we have performed an unbiased genetic screen to identify genes that when overproduced promote anchorage-independent cell growth in a human mammary epithelial cell (TL-HMEC) line. Our screen identified a cell adhesion molecule PVRL4 (poliovirus receptor-like 4), also known as Nectin-4, which

we demonstrate to be a potent mediator of anchorage-independent colony formation in normal epithelial and cancer cells alike. We demonstrate that PVRL4 promotes the attachment of individual cells to each other via engaging its receptor PVRL1 on a juxtaposed cell. PVRL4-mediated cell-to-cell attachment triggers integrin β4 signaling in a matrix attachment-independent manner, and interfering with this pathway blocks PVRL4-driven anchorage-independence. Our findings point to a model in which signaling via PVRL4-PVRL1-driven cell–cell contacts serves as a surrogate for cell–matrix signaling in conditions of anchorage loss, thus enabling anoikis evasion and subsequent cellular expansion.

Finally, we show that blocking PVRL4-driven cell–cell contact assembly with monoclonal antibodies potently inhibits anchorage-independent cellular expansion in vitro as well as the growth of orthotopically implanted tumors in vivo, thus validating the therapeutic utility of this approach.

## Results

### A genetic screen for genes that enable anchorage-independence

To identify novel genes involved in the transformation of epithelial cells, we carried out a genetic screen for colony formation in the absence of substratum attachment using a TL-HMEC in vitro transformation system. In this model, hTERT-immortalized human mammary epithelial cells are transduced with SV40 Large T antigen (*Zhao et al., 2003*). TL-HMECs cannot survive and proliferate in a semi-solid medium, such as methylcellulose, in the absence of substratum attachment; however, ectopic expression of oncogenic H-RAS$^{V12}$, myristoylated PI3K catalytic subunit (*Zhao et al., 2003*), or shRNA against PTEN (*Westbrook et al., 2005*) enables anchorage-independent cellular expansion and formation of macroscopic colonies. We screened a retrovirus-encoded library of 8000 human open reading frames (ORFeome 1.1) (*Rual et al., 2004*) (*Figure 1A*) and individually recovered stably integrated ORFs from 732 resulting macroscopic colonies by PCR, followed by determination of the ORF identity by sequencing. Forty ORFs that were identified in two independent screen replicates were individually validated to confirm the anchorage-independent growth phenotype (*Figure 1B*, see also *Supplementary file 1A*). Eleven ORFs potently induced anchorage-independent colony formation (a fivefold increase over background), and another eight ORFs produced a moderate phenotype (a two- to fivefold increase over background). According to the recently published copy number variation analysis across a variety of tumor types (*Beroukhim et al., 2010*), eight out of 11 ORFs that potently induced anchorage-independent growth localized to statistically defined peaks of focal amplification in at least one tumor subtype (p=0.009, Fisher's exact test), providing strong evidence for positive selection of these genes in cancer (*Supplementary file 1B*). In addition, ORFs corresponding to two genes previously shown to promote anchorage-independent colony formation were identified in our screen. Ectopic expression of SULF2 (sulfatase 2) had been previously found to induce anchorage-independent growth in human bronchial epithelial cells (*Lemjabbar-Alaoui et al., 2010*) and NPM1 (nucleophosmin 1) overexpression had been previously found to enable NIH 3T3 fibroblast colony formation in methylcellulose (*Kondo et al., 1997*).

We focused our attention on the top scoring candidate identified in the screen, PVRL4 (poliovirus receptor-like 4). Ectopic expression of PVRL4 strongly induced anchorage-independent colony formation (*Figure 1B*), whereas its endogenous expression was not detectable in TL-HMECs (*Figure 2B*), consistent with the documented absence of its expression in normal mammary gland (*Fabre-Lafay et al., 2007*).

### The cytoplasmic region of PVRL4 is not required for anchorage-independent growth

Recently identified to be a receptor for measles virus, PVRL4 (*Muhlebach et al., 2011*; *Noyce et al., 2011*) belongs to a family of poliovirus receptor-related transmembrane proteins, also known as nectins, which localize to sites of cell junctions in various cell types (reviewed in *Takai et al., 2008*). Similarly to other nectins, PVRL4 contains an extracellular region with three immunoglobulin-like domains, a single transmembrane domain, and a short cytoplasmic tail (*Reymond et al., 2001*). Extracellular regions of nectins engage in *trans*-interactions on the interface of two juxtaposed cells. Through a stretch of four C-terminal amino acids, *trans*-interacting nectins recruit afadin, a scaffold-like mediator of downstream signaling, triggering assembly of the afadin/Rap1/p120$^{ctn}$ complex, which positively regulates membrane retention of E-cadherin, assisting in adherens junction (AJ) formation in MDCK cells (*Hoshino et al., 2005*).

To identify regions of PVRL4 responsible for driving anchorage-independent growth, we created a series of PVRL4 deletion constructs (*Figure 2A,B*, see also *Supplementary file 1C*). Remarkably,

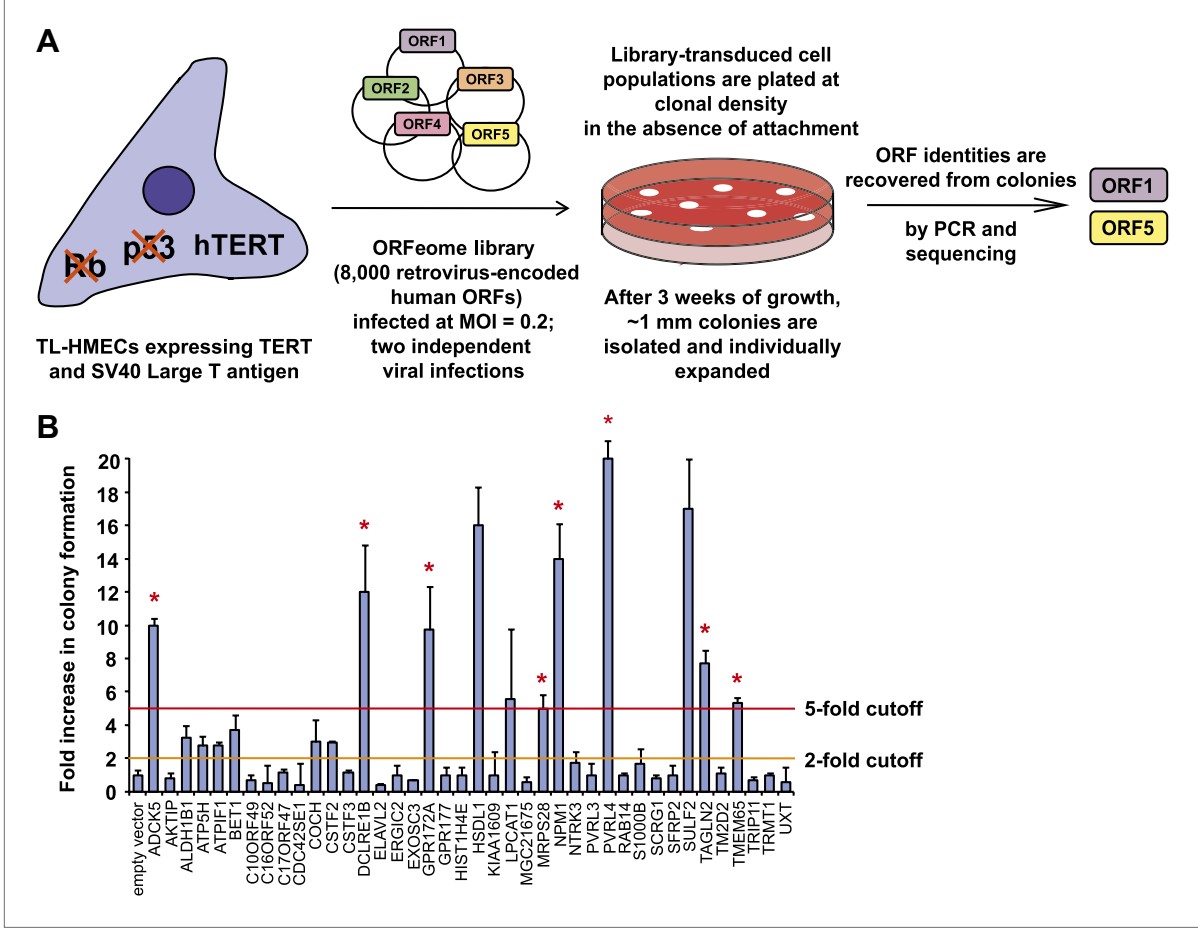

**Figure 1**. A genetic screen for drivers of anchorage-independent growth in human mammary epithelial cells. (**A**) A schematic of the screen. TL-HMECs were transduced with the ORFeome library (8000 CMV promoter-driven human open reading frames [ORFs] in a retroviral vector) at a multiplicity of infection (MOI) of 0.2 and plated into semi-solid medium. Macroscopic colonies were isolated, individually expanded, and the identities of ORF inserts were determined by sequencing. (**B**) ORFs recovered from two independent screen replicates were individually transduced into TL-HMECs and plated into semi-solid medium. Colonies were counted and colony numbers were normalized to an empty vector-transduced sample. Asterisks denote strongly validated ORFs that localize to focal amplification peaks in at least one tumor subtype. Assays were performed in triplicate (error bars ± SD).

deletion of the entire cytoplasmic region of PVRL4 did not affect its ability to promote TL-HMEC colony formation (***Figure 2C***) or cell viability (***Figure 2D***) in the absence of substratum attachment.

Mammary epithelial cells are known to undergo an alternative anoikis-related cell death program characterized by robust induction of transcripts associated with terminal squamous differentiation when apoptosis is suppressed (***Mailleux et al., 2007***). TL-HMECs in particular do not initiate classic apoptotic responses in response to the loss of substratum attachment, likely due to the anti-apoptotic action of Large T antigen (not shown), but instead potently upregulate multiple squamous differentiation markers, including TGM1 (transglutaminase 1), KRT6A (keratin 6), and IVL (involucrin), and we found that both full-length PVRL4 and its cytoplasmic deletion mutant reduced this effect (***Figure 2E,F***). Taken together, these data demonstrate that PVRL4 protects TL-HMECs from the differentiation associated with anchorage loss and promotes subsequent cellular expansion. Importantly, this phenotype is facilitated by the extracellular part of PVRL4 rather than by its intracellular region.

## PVRL4 facilitates a cell–cell contact assembly that is required for anchorage-independent growth

Consistent with its role in promoting cell–cell contact formation, PVRL4 drives rapid association of TL-HMECs into multicellular clusters in suspension (***Figure 3A,B***). We sought to rule out the possibility that PVRL4-induced colonies were due to an artifact associated with either incomplete dissociation of

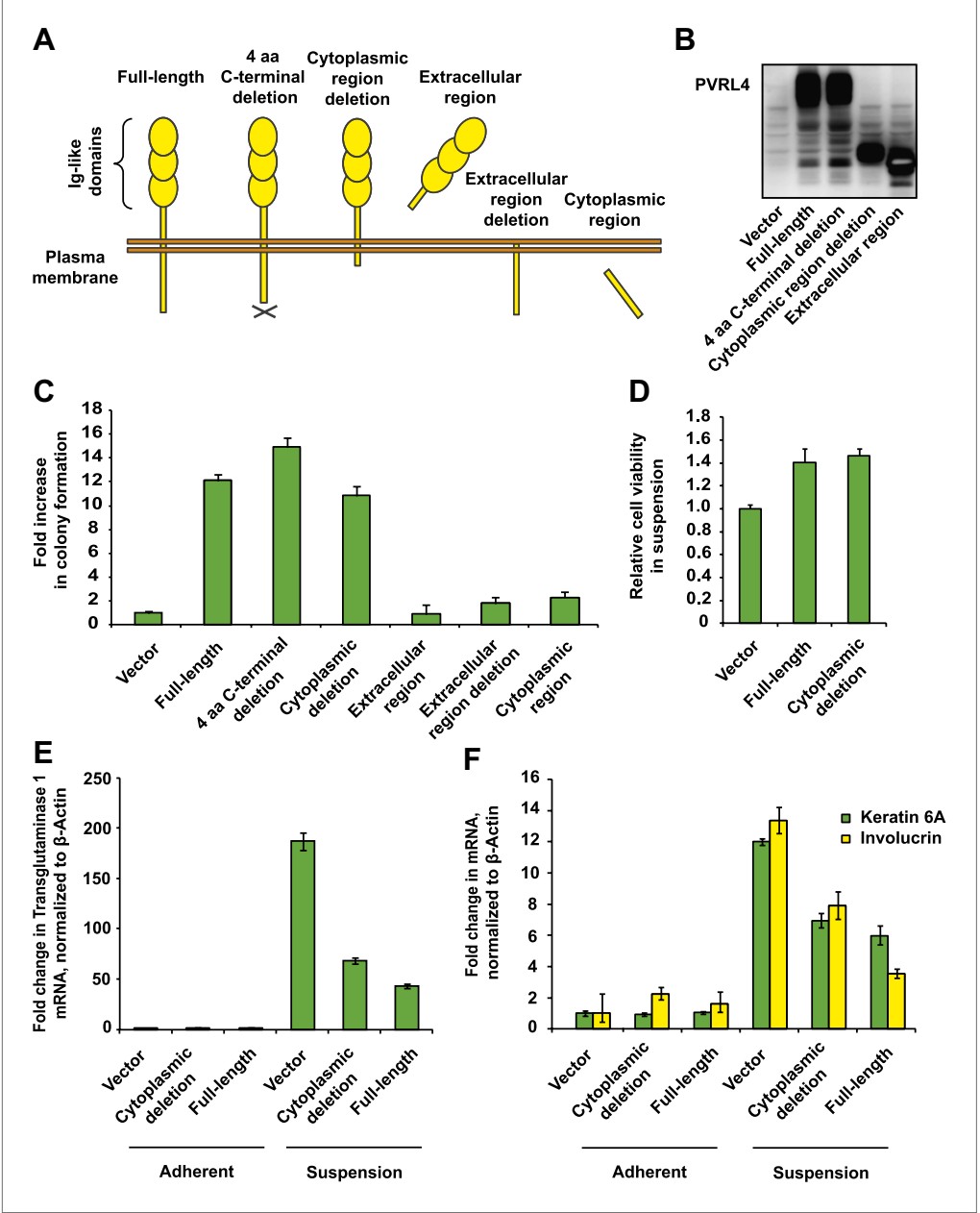

**Figure 2**. PVRL4-induced anchorage-independent colony formation is carried out through its extracellular region. (**A**) and (**B**) A series of PVRL4 deletion constructs were designed and their expression confirmed by Western blot. (**C**) PVRL4 mutants from (**A**) were tested for their ability to induce anchorage-independent colony formation in triplicate (error bars ± SD). (**D**) Cells with full-length PVRL4 or the cytoplasmic region deletion mutant were assayed for viability under conditions of anchorage deprivation by measuring total ATP content in cells cultured on ultra-low attachment plates for 72 hr. Values were normalized to an empty vector-transduced sample. Assays were performed in triplicate (error bars ± SD). (**E**) and (**F**) TL-HMECs expressing empty vector, full-length PVRL4 or cytoplasmic region deletion mutant containing cells were cultured on tissue culture-treated (adherent) or ultra-low attachment (suspension) dishes for 72 hr. RNA was isolated and mRNA levels for TGM1 (**E**) and KRT6A and IVL (**F**) were measured by RT-qPCR. Transcript levels were normalized to β-actin. qPCR was performed in quadruplicate (error bars ± SD).

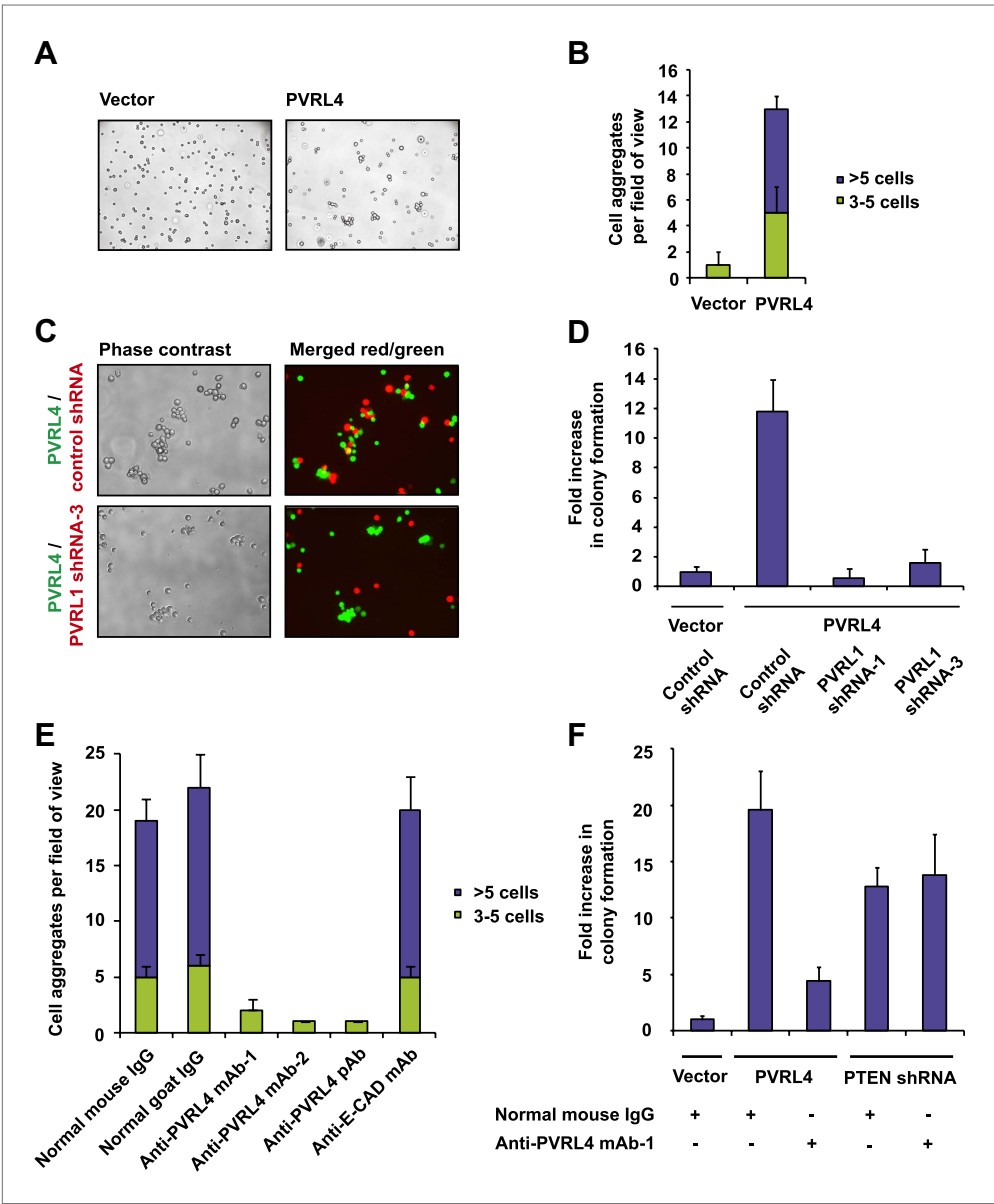

**Figure 3**. PVRL4 facilitates cell-to-cell attachment, inhibition of which suppresses anchorage-independence. (**A**) and (**B**) PVRL4 promotes cell clustering of TL-HMECs. Cells were dissociated off the tissue culture surface with trypsin-free cell dissociation buffer and kept in suspension for 1 hr. Small (3–5 cells) and large (>5 cells) cell clusters per field of view were counted, n = 3 (error bars ± SD). (**C**) GFP-labeled PVRL4-expressing TL-HMECs were allowed to aggregate with dsRed-labeled cells expressing either a PVRL1-targeting shRNA or a control shRNA. Representative phase-contrast and fluorescent images (red and green channels superimposed) are shown. (**D**) PVRL4 was co-expressed with the indicated shRNAs and anchorage-independent colony formation in TL-HMECs was assayed. Values were normalized to an empty vector-transduced sample. Assays were performed in triplicate (error bars ± SD). (**E**) PVRL4-expressing TL-HMECs were assayed for clustering in the presence of the indicated antibodies or isotype controls. Cell clusters were quantified as before. (**F**) Anchorage-independent growth induced by PVRL4 or an shRNA against PTEN was assayed in the presence of PVRL4-targeting antibody or control IgG. Colony numbers were normalized to the control sample. Anchorage-independent colony formation assays were performed in triplicate (error bars ± SD).

The following figure supplements are available for figure 3:

**Figure supplement 1**. PVRL4-driven anchorage-independent colonies originate from single cells.

*Figure 3. Continued on next page*

*Figure 3. Continued*

**Figure supplement 2**. Potential preformed clusters of TL-HMECs do not contribute to anchorage-independent colony numbers.

**Figure supplement 3**. Depletion efficiency of individual anti-PVRL1 shRNAs.

**Figure supplement 4**. Anti-PVRL4 antibodies block PVRL4-driven cell–cell clustering.

**Figure supplement 5**. PVRL4-driven cell–cell clustering is inhibited by antibodies against PVRL1.

cells prior to seeding into methylcellulose or potential de novo association of cells during growth in anchorage-independent conditions. To address this, we mixed equal numbers of dsRed- and GFP-labeled TL-HMECs expressing PVRL4 and (i) seeded them into methylcellulose or (ii) co-cultured the mixed population on an adherent surface, subsequently seeding them into methylcellulose. Examining the colors of resulting colonies revealed that out of 56 colonies from sample (i), all 56 were single-color colonies, whereas in sample (ii), 39 out of 40 colonies were single-color, and only one colony contained both GFP and dsRed-positive cells (*Figure 3—figure supplement 1*). The rarity of double-color colonies demonstrates that PVRL4-induced colonies originate from individual cells. Consistent with this observation, passing cells through a 35 μm cell strainer immediately prior to being seeded in the absence of anchorage did not affect colony numbers (*Figure 3—figure supplement 2*), further confirming that PVRL4-induced colonies are clonal in origin and that the observed increase in colony numbers is not a result of pre-existing multicellular clusters having a survival advantage over single cells in the absence of anchorage.

The extracellular region of PVRL4 exhibits preferential affinity for *trans*-interaction with PVRL1 (*Reymond et al., 2001*), a cell adhesion molecule endogenously expressed in TL-HMECs. To test whether PVRL1 was necessary for PVRL4-driven cell-to-cell attachment, we used an shRNA to stably deplete the PVRL1 transcript in a dsRed-labeled population of TL-HMECs and allowed cells to aggregate with PVRL4-expressing GFP-labeled TL-HMECs. PVRL1 depletion resulted in exclusion of dsRed-labeled cells from multicellular clusters, whereas control shRNA-expressing cells were readily incorporated (*Figure 3C*), suggesting that PVRL4-mediated cell–cell contacts are carried out through interaction with its receptor PVRL1.

We then asked whether PVRL1 was similarly required for PVRL4-driven anchorage-independence. Indeed, stable depletion of the PVRL1 transcript by two independent shRNA constructs (*Figure 3—figure supplement 3*) abolished anchorage-independent colony formation (*Figure 3D*), paralleling the effect of PVRL1 depletion on PVRL4-driven cell-to-cell attachment.

The above data raise the possibility that PVRL4-PVRL1 *trans*-interaction on juxtaposed cells is mechanistically involved in driving anchorage-independent growth. Therefore, we sought a physical means by which to disrupt PVRL4-induced cell-to-cell attachment to examine its effects on colony formation. We asked whether antibodies directed towards the extracellular domain of PVRL4 could block PVRL4-induced cluster formation and anchorage-independent growth. Cell clustering was completely abrogated in the presence of three independent PVRL4 antibodies, whereas control IgG or a blocking antibody against E-cadherin (DECMA-1) did not produce such an effect (*Figure 3E*, *Figure 3—figure supplement 4*). Similarly, an antibody targeting the extracellular region of PVRL1 inhibited PVRL4-induced cell clustering (*Figure 3—figure supplement 5*). PVRL4-driven colony formation was inhibited to almost a basal level in the presence of a monoclonal antibody targeting the extracellular region of PVRL4, but no such effect was observed on colonies induced by PTEN shRNA (*Figure 3F*), demonstrating that the observed inhibition is due to a direct effect of an antibody and not due to its general toxicity. Our data indicate that PVRL4 enables anchorage-independent growth in a manner which is dependent on PVRL4-PVRL1-driven cell-to-cell attachment, suggesting that these intercellular contacts may be providing a survival benefit in the context of altered matrix anchorage.

## The transmembrane and cytoplasmic regions of both PVRL4 and PVRL1 are dispensable for cell-to-cell attachment and anchorage-independent growth

To further probe the functional link between PVRL4-driven cell-to-cell attachment and anchorage-independent growth, we asked whether the PVRL4-PVRL1-mediated cell surface interaction alone is

sufficient for this phenotype. To address this question, we created chimeric constructs in which extracellular regions of PVRL1 and PVRL4 were fused to the transmembrane regions of an unrelated transmembrane molecule, CD8 (*Figure 4A*), and the cytoplasmic regions of both molecules were deleted. We first introduced PVRL4-CD8tm into TL-HMECs, concomitantly depleting endogenous PVRL1 by RNAi (*Figure 4D*). Membrane localization of the chimeric construct was verified by immunofluorescence (*Figure 4—figure supplement 1*). Expression of PVRL4-CD8tm induced both cell clustering and anchorage-independent colony formation, and both phenotypes were suppressed by stable depletion of endogenous PVRL1 (*Figure 4B,C*). Importantly, clustering was restored when two populations of clustering-incompetent cells, PVRL4-CD8tm expressing, PVRL1-depleted cells (PVRL4$^+$; PVRL1$^-$) and control cells (PVRL4$^-$; PVRL1$^+$), were mixed together, independently verifying that cell-to-cell attachment is mediated by the PVRL4-PVRL1 *trans*-interacting module (*Figure 4—figure supplement 2*). Both cell clustering and colony formation defects induced by PVRL1 shRNA were rescued by the expression of the shRNA-resistant PVRL1-CD8tm construct. Taken together, these data indicate an on-target nature of the PVRL1 depletion phenotype and demonstrate that ability to withstand the absence of anchorage is driven by the PVRL4-PVRL1 cell-surface *trans*-interaction and not by intracellular or transmembrane regions of either molecule.

## Multiple oncogenic perturbations induce cell clustering

We have shown that PVRL4-driven anchorage-independent growth is dependent on its involvement in driving cell-to-cell attachment, which raises an interesting possibility that in the context of the altered matrix environment that tumor cells encounter, certain types of intercellular interactions function to enable anchorage-independent survival. Along these lines, we hypothesized that other oncogenic perturbations that drive anchorage-independent growth (*Figure 4—figure supplement 3*) may also promote cell-to-cell attachment as a part of their survival strategy. Either expression of mutant Ras$^{V12}$, or myristoylated PI3K catalytic subunit (*Figure 4—figure supplement 4*), or shRNA-mediated depletion of PTEN (*Figure 4—figure supplement 5*; for depletion efficiencies see *Figure 4—figure supplement 6*) triggered multicellular cluster formation in TL-HMECs. None of the perturbations induced endogenous PVRL4 expression, as measured by FACS (not shown), which suggests that clustering in these cases was mediated through other adhesion mechanisms. Mixing PTEN shRNA-expressing GFP-labeled cells with control shRNA-expressing dsRed-labeled cells triggered the formation of multicellular clusters that contained both GFP- and dsRed-labeled cells (*Figure 4—figure supplement 7*), demonstrating that, in a manner similar to PVRL4, depletion of PTEN enabled attachment of TL-HMECs to cells that do not carry the perturbation as well as those that do. Taken together, these observations suggest that increased cell-to-cell attachment is a phenotype that multiple oncogenic perturbations converge upon.

## Integrin β4 physically interacts with PVRL4 and is necessary for the PVRL4-driven anchorage-independence

We next sought to gain insight into the mechanism by which PVRL4-PVRL1 cell surface *trans*-interaction becomes translated into survival advantage in conditions of anchorage loss. Neither of the two molecules possesses catalytic activity; in addition, our data show that transmembrane as well as cytoplasmic regions are dispensable for anchorage-independence. Therefore, we considered the possibility that the assembly of a PVRL4-PVRL1 interacting module on the interface of two neighboring cells may be triggering lateral recruitment and/or activation of cell surface-localized proteins, which, in turn, conveys a prosurvival signal. To identify potential cell surface-localized binding partners of PVRL4, we created a C-terminally HA/FLAG-tagged PVRL4 construct and first verified that cell–cell clustering as well as anchorage-independent growth phenotypes were not affected by the addition of a C-terminal tag (not shown). We then performed immunoprecipitations with anti-HA agarose beads from lysates of TL-HMECs induced to express either HA/FLAG-tagged PVRL4 or HA/FLAG-tagged GFP and subjected the eluates to tandem mass spectrometry analysis. Since only the extracellular regions of PVRL4 and PVRL1 are required for anchorage-independent colony formation, we searched the list of PVRL4 IP-specific peptides for those that had at least three unique peptides corresponding to cell surface-localized proteins, as classified by Gene Ontology. We identified the transmembrane protein integrin β4 as specifically interacting with HA/FLAG-PVRL4 (*Figure 5A*). To validate the putative PVRL4–integrin β4 interaction directly in cells under conditions of anchorage deprivation, we performed immunoprecipitations with anti-HA beads from lysates of suspension-incubated TL-HMECs that expressed either

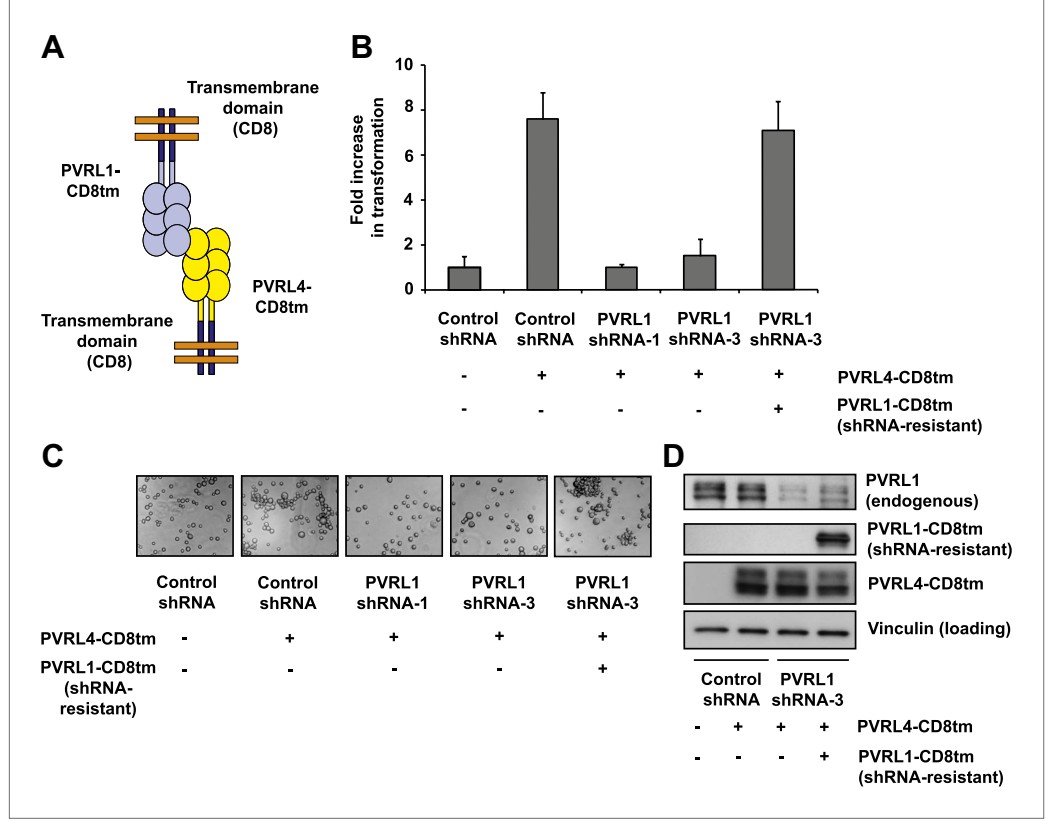

**Figure 4**. Expression of extracellular regions of PVRL4 and PVRL1 on the cell surface is sufficient for anchorage-independence. (**A**) Schematics of chimeric constructs containing extracellular domains of PVRL4 or an shRNA-resistant version of PVRL1 fused to the transmembrane domain of CD8 (blue). (**B**) and (**C**) TL-HMECs were stably transduced with the indicated combinations of expression constructs and assayed for anchorage-independent growth (**B**) and clustering (**C**). Colony numbers were normalized to the control sample. Anchorage-independent colony formation assays were performed in triplicate (error bars ± SD). (**D**) Expression levels of endogenous and chimeric proteins were verified by Western blot.

The following figure supplements are available for figure 4:

**Figure supplement 1**. Plasma membrane localization of PVRL4-CD8tm construct in TL-HMECs.

**Figure supplement 2**. Cell–cell clustering is driven by PVRL4-PVRL1 trans-interactions between individual cells.

**Figure supplement 3**. Expression of shRNA constructs against PTEN or constitutively active mutants of RAS and PI3K induces anchorage-independent growth.

**Figure supplement 4**. Constitutively active mutants of RAS and PI3K induce cell–cell clustering.

**Figure supplement 5**. PTEN depletion induces cell–cell clustering.

**Figure supplement 6**. Depletion efficiency of individual anti-PTEN shRNAs.

**Figure supplement 7**. Cell–cell clustering induced by depletion of PTEN is heterotypic.

HA/FLAG-tagged PVRL4 or HA/FLAG-tagged GFP. Immunoblotting for integrin β4 verified a specific association of integrin β4 with PVRL4 (**Figure 5B**). We next asked if the integrin β4–PVRL4 association was affected by cell clustering. The same amount of integrin β4 coprecipitated with HA/FLAG-tagged PVRL4 from lysates prepared from multicellular clusters and from single-cell suspension. Similarly, the amount of coprecipitating integrin β4 was not affected by RNAi-mediated depletion of PVRL1

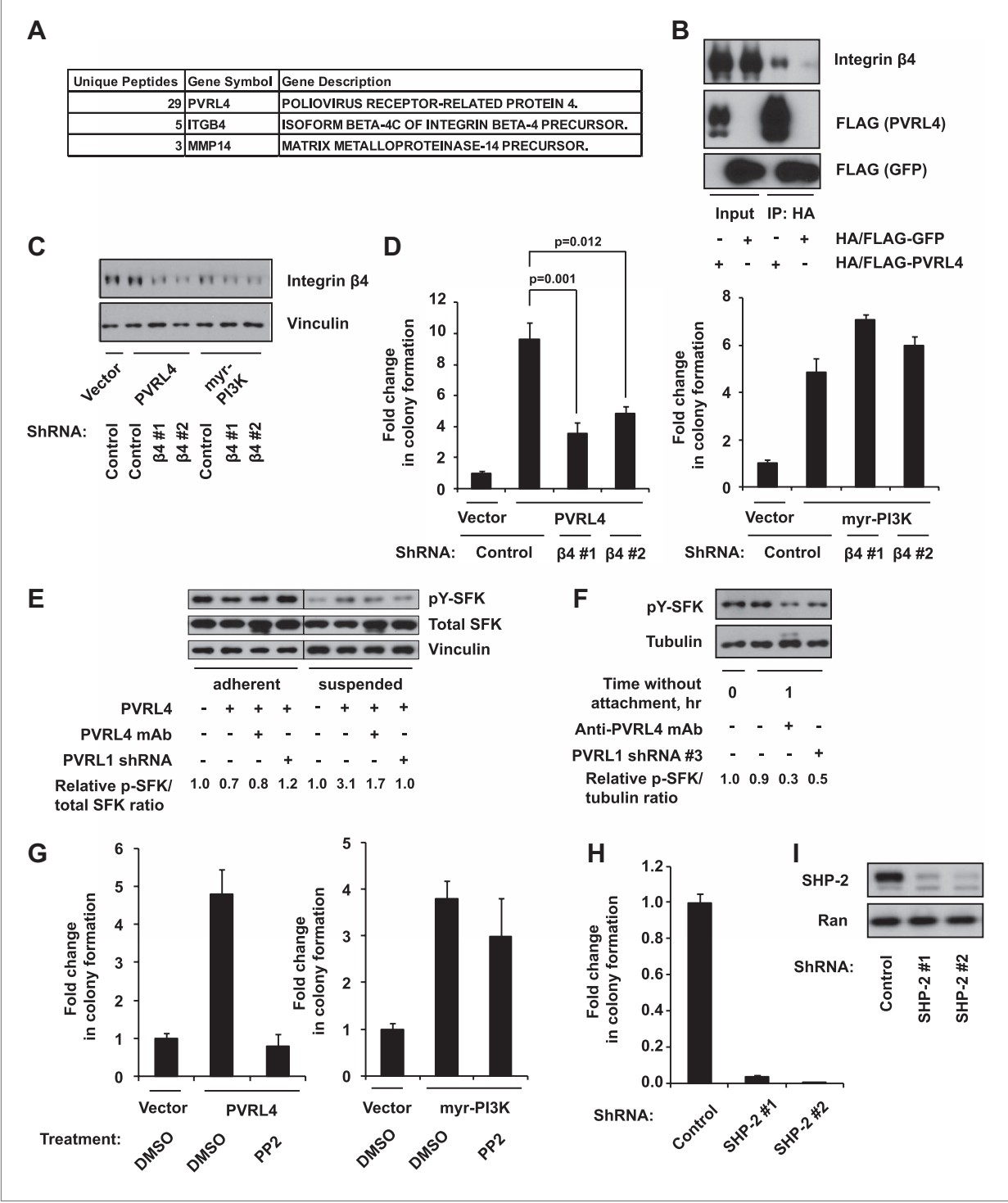

**Figure 5**. PVRL4-driven cell-to-cell attachment promotes anchorage-independence via integrin β4-associated signaling. (**A**) Cell surface-localized proteins interacting with HA/FLAG-tagged PVRL4, but not with HA/FLAG-tagged GFP, as determined by mass spectrometry. (**B**) TL-HMECs expressing HA/FLAG tagged PVRL4 or HA/FLAG-tagged GFP were detached from the adherent surface with the enzyme-free cell dissociation buffer and incubated in suspension for 1 hr. Immunoprecipitations were performed with HA beads, followed by Western blot with FLAG and integrin β4 antibody. (**C**) TL-HMECs expressing vector control, PVRL4, or myr-PI3K were stably transduced with the indicated shRNA constructs and integrin β4 levels were assayed by Western blot. (**D**) TL-HMECs from (**C**) were assayed for anchorage-independent colony formation. Colony numbers were normalized to the vector control sample. Assays were performed in triplicate (error bars ± SD). (**E**) TL-HMECs stably transduced with the indicated constructs were

*Figure 5. Continued on next page*

*Figure 5. Continued*

detached from the adherent surface with enzyme-free cell dissociation buffer and incubated in 0.5% methylcellulose in suspension for 6 hr or cultured on an adherent surface for 48 hr. Levels of pY416-SFK (Src family kinases), total SFK, and vinculin loading control were measured by Western blot. Band intensity was measured with ImageJ software. (**F**) PVRL4-expressing TL-HMEC cells transduced with control or anti-PVRL1 shRNA were incubated in suspension in the conditions indicated. Levels of pY416-SFK (Src family kinases) and tubulin loading control were measured by Western blot. Band intensity was measured with ImageJ software. (**G**) TL-HMECs stably expressing PVRL4 or control vector were assayed for anchorage-independent colony formation in the presence of PP2 or vehicle control. Colony numbers were normalized to the vector sample. Assays were performed in triplicate (error bars ± SD). (**H**) TL-HMECs expressing PVRL4 were stably transduced with the indicated shRNA constructs and assayed for anchorage-independent colony formation. Colony numbers were normalized to the vector control sample. Assays were performed in triplicate (error bars ± SD). (**I**) SHP-2 levels were assayed by Western blot in TL-HMEC lysates from (**H**).

The following figure supplements are available for figure 5:

**Figure supplement 1**. Interaction of PVRL4 with integrin β4.

(*Figure 5—figure supplement 1A*). These data point to a *cis* mode of association between PVRL4 and integrin β4 on the membrane of the same cell.

We took further steps to investigate the interaction between integrin β4 and PVRL4, asking whether deletion of the cytoplasmic region of PVRL4 will disrupt its association with integrin β4. Immunoprecipitation of integrin β4 from TL-HMECs which were induced to express either a full-length or a cytoplasmic deletion version of PVRL4 revealed that both full-length and mutant versions of PVRL4 coprecipitated with integrin β4, and not with a control IgG (*Figure 5—figure supplement 1B*). This finding parallels our earlier demonstration that the cytoplasmic region of PVRL4 is dispensable for the anchorage-independence phenotype, prompting us to investigate a potential mechanistic link between integrin β4 and the PVRL4-driven anchorage-independence phenotype.

To directly test the involvement of integrin β4 in PVRL4-driven anchorage-independence, we used two independent shRNA constructs for stable depletion of integrin β4 from TL-HMECs (*Figure 5C*). Depletion of integrin β4 had no effect on TL-HMEC attachment to the tissue culture vessel or on proliferation in adherent conditions; neither did it affect PVRL4-driven cell clustering (not shown). However, PVRL4-induced anchorage-independent colony formation was markedly reduced in the presence of integrin β4-specific shRNAs (*Figure 5D*). Importantly, reliance on integrin β4 was specific to colony growth promoted by PVRL4, as PI3K-driven colony numbers were not affected by its depletion.

## PVRL4-driven cell-to-cell attachment enables sustained Src family kinase activation in conditions of anchorage deprivation

A unique member of the integrin family, integrin β4 contains an atypically long C-terminal region (1017 amino acids) which does not take part in canonical focal adhesion formation, but instead, creates attachment points for keratin-containing intermediate filaments in adherent cells, forming hemidesmosomes (*Giancotti, 2007*). In contrast to other integrins, the C-terminus of integrin β4 does not bind FAK. However, it can activate Src family kinases (SFKs) in a FAK-independent manner via recruitment of SHP-2 phosphatase.

In substratum-attached cells, ligation of integrins to the matrix maintains a constitutive level of active SFKs, whereas loss of anchorage inactivates SFKs. To test whether cells in PVRL4-induced clusters are able to maintain SFK activity in conditions of anchorage loss, we measured levels of autophosphorylated SFKs in TL-HMECs after 6 hr incubation in suspension (*Figure 5E*). Levels of activated SFK dropped precipitously in control cells while remaining high in PVRL4-expressing cells but not in PVRL4-expressing, PVRL1-depleted cells. Moreover, the addition of anti-PVRL4 antibody also reduced the level of SFK autophosphorylation in suspension. A similar effect was observed after a short-term incubation of PVRL4-expressing TL-HMECs in suspension (*Figure 5F*), where levels of SFK activation observed in freshly detached cells (0 hr time point) were maintained in cells that were allowed to attach to each other but not in cells in which clustering was prevented by either anti-PVRL4 antibody or PVRL1 depletion.

Having shown that PVRL4-driven cell-to-cell attachment could preserve SFK activation even in the absence of substratum attachment, we hypothesized that SFK activation is a critical driver of the observed ability of PVRL4-expressing cells to withstand the loss of anchorage. In support of this, PVRL4-driven but not PI3K-driven anchorage-independent colony formation was abrogated in the presence of PP2, a chemical inhibitor of SFK activity (*Figure 5G*).

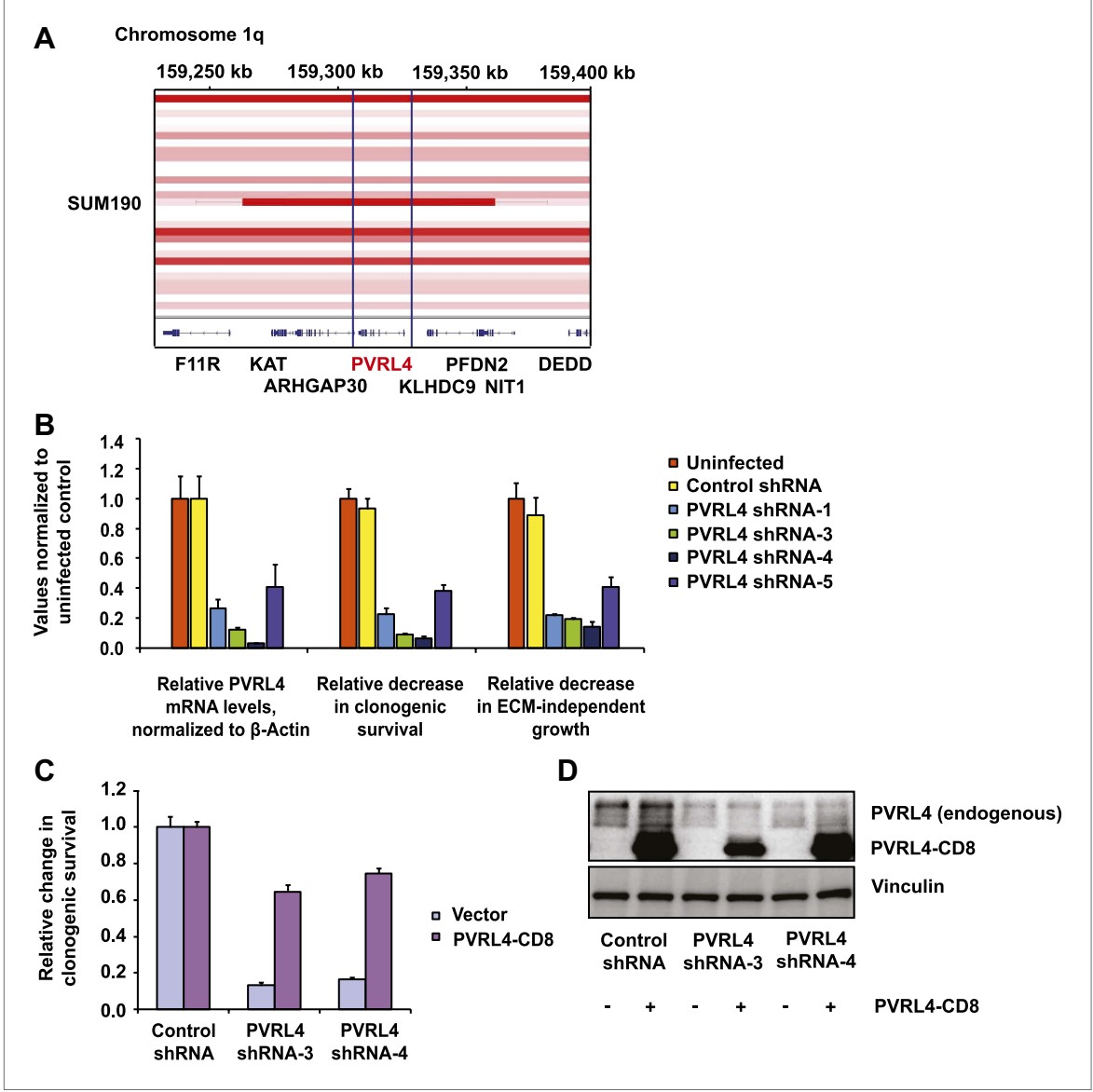

**Figure 6**. PVRL4 is amplified in breast cancer and is essential for the transformed phenotype of cancer cells. (**A**) A view from the integrated Genome Viewer program showing focal amplification of the PVRL4 locus in SUM190 cells. The degree of amplification is denoted by the intensity of the color. (**B**) PVRL4 mRNA was stably depleted from SUM190 cells by four independent shRNAs. Transcript levels were measured by RT-qPCR and normalized to β-actin. qPCR was performed in quadruplicate (error bars ± SD). PVRL4-depleted and control cells were assayed for clonogenic survival and anchorage-independent colony formation. Assays were performed in triplicate (error bars ± SD). All values were normalized to the uninfected control sample. ECM: extracellular matrix. (**C**) The PVRL4-CD8 chimeric construct was used to rescue the defect in clonogenic survival observed with RNAi-mediated PVRL4 depletion. Assays were performed in triplicate (error bars ± SD). Colony numbers were normalized to the control shRNA sample. (**D**) Expression levels of endogenous and chimeric proteins were verified by Western blot.

The following figure supplements are available for figure 6:

**Figure supplement 1**. PVRL4 induces clustering of SUM190 cells which is blocked by antibodies against PVRL4.

**Figure supplement 2**. PVRL4 induces clustering of SUM190 cells which is blocked by RNAi against PVRL4.

**Figure supplement 3**. PVRL4 induces attachment of SUM190 cells to microvascular endothelial cells which is blocked by antibodies against PVRL4.

**Figure supplement 4**. PVRL4 induces attachment of SUM190 cells to microvascular endothelial cells which is blocked by RNAi against PVRL4.

*Figure 6. Continued on next page*

*Figure 6. Continued*

**Figure supplement 5**. Stable depletion of PVRL4 transcript in BT-474 and Sk-BR-3 cell lines.

**Figure supplement 6**. PVRL4 depletion affects clonogenic growth of BT-474 and Sk-BR-3 cell lines.

**Figure supplement 7**. PVRL4 depletion affects anchorage-independent growth of BT-474 and Sk-BR-3 cell lines.

Activation of SFKs by integrin β4 is carried out via recruitment of SHP-2 phosphatase. Since PVRL4-driven anchorage-independent growth is inhibited both by integrin β4 depletion and by chemical inhibition of SFK activity, we reasoned that depletion of SHP-2 might have a similar effect. Indeed, two independent shRNAs against SHP-2 completely abrogated colony formation induced by PVRL4 (***Figure 5H,I***). Taken together, these data suggest that PVRL4-driven anchorage-independent growth is facilitated via activation of the integrin β4/SHP-2/Src signaling pathway in a manner that is dependent on cell-to-cell attachment.

## PVRL4 maintains the transformed properties of breast cancer cells in vitro

Having uncovered a role for PVRL4 in enabling epithelial cells to escape growth restriction associated with the lack of proper matrix anchorage, we next sought to test its involvement in driving the tumorigenic properties of breast cancer cells. We first focused on an inflammatory breast tumor cell line SUM190, which contains a particularly high-level focal amplification (~50 kb) of the *PVRL4* genomic locus (***Figure 6A***) (***Beroukhim et al., 2010***). To ask whether PVRL4 is involved in enabling anchorage-independent growth of these cells, we designed four independent shRNAs targeting PVRL4 and stably expressed them in SUM190 cells, confirming the extent of depletion by RT-qPCR (***Figure 6B***). Paralleling effects seen in TL-HMECs, spontaneous self-clustering of SUM190 cells was abrogated by anti-PVRL4 antibodies (***Figure 6—figure supplement 1***) or by RNAi against PVRL4 (***Figure 6—figure supplement 2***). Similarly, heterotypic attachment of SUM190 cells with human lung microvascular endothelial (HMVEC-L) cells was prevented by anti-PVRL4 antibodies (***Figure 6—figure supplement 3***) and by RNAi (***Figure 6—figure supplement 4***). RNAi against PVRL4 potently reduced both anchorage-independent colony formation and clonogenic growth of SUM190 cells, indicating that PVRL4 is involved in the growth and survival of breast cancer cells. Importantly, observed effects correlated with the degree of RNAi-mediated mRNA depletion (***Figure 6B***). To verify that the effects of PVRL4 depletion are not limited to SUM190 cells, we expressed anti-PVRL4 shRNAs in two additional breast cancer cell lines, Sk-BR-3 and BT-474 (***Figure 6—figure supplement 5***), which both express lower levels of PVRL4 than SUM190 cells (***Fabre-Lafay et al., 2007***), and observed a strong defect in clonogenic growth (***Figure 6—figure supplement 6***) and reduced proliferation in the absence of anchorage (***Figure 6—figure supplement 7***).

To confirm that the observed phenotype was specific for PVRL4 depletion, we used a PVRL4-CD8tm construct to rescue the colony formation defect in SUM190 cells. Coexpression of PVRL4-CD8tm in the presence of PVRL4-targeting shRNAs alleviated the clonogenic survival defect, confirming its on-target nature (***Figure 6C,D***). This result also demonstrates that the extracellular region of PVRL4 alone is sufficient for potentiating clonogenic survival of breast cancer cells, consistent with PVRL4-CD8tm ability to enable anchorage-independent colony formation in TL-HMECs.

## Targeting PVRL4 inhibits tumor growth in vivo

To determine whether colony growth inhibition induced by PVRL4 depletion in vitro was also relevant for in vivo tumor growth, we stably expressed either control shRNA or an anti-PVRL4 shRNA in a panel of breast cancer cell lines and orthotopically implanted them into the mammary fat pads of nude mice. Depletion of PVRL4 by RNAi from SUM190 (***Figure 7A,B***), SUM185 (***Figure 7C***), and BT-474 cells (***Figure 7—figure supplement 1***) inhibited xenograft growth, verifying the importance of PVRL4 to cancer cell growth in vivo as well as in vitro.

Having previously shown that blocking PVRL4-driven cell-to-cell attachment with a monoclonal antibody suppresses SFK activation and anchorage-independent growth of TL-HMECs in vitro, we sought to determine whether this antibody would similarly inhibit SUM190 xenograft growth in an in vivo setting. To address this, we treated mice bearing ~50 mm³ SUM190-eGFP xenografts with four

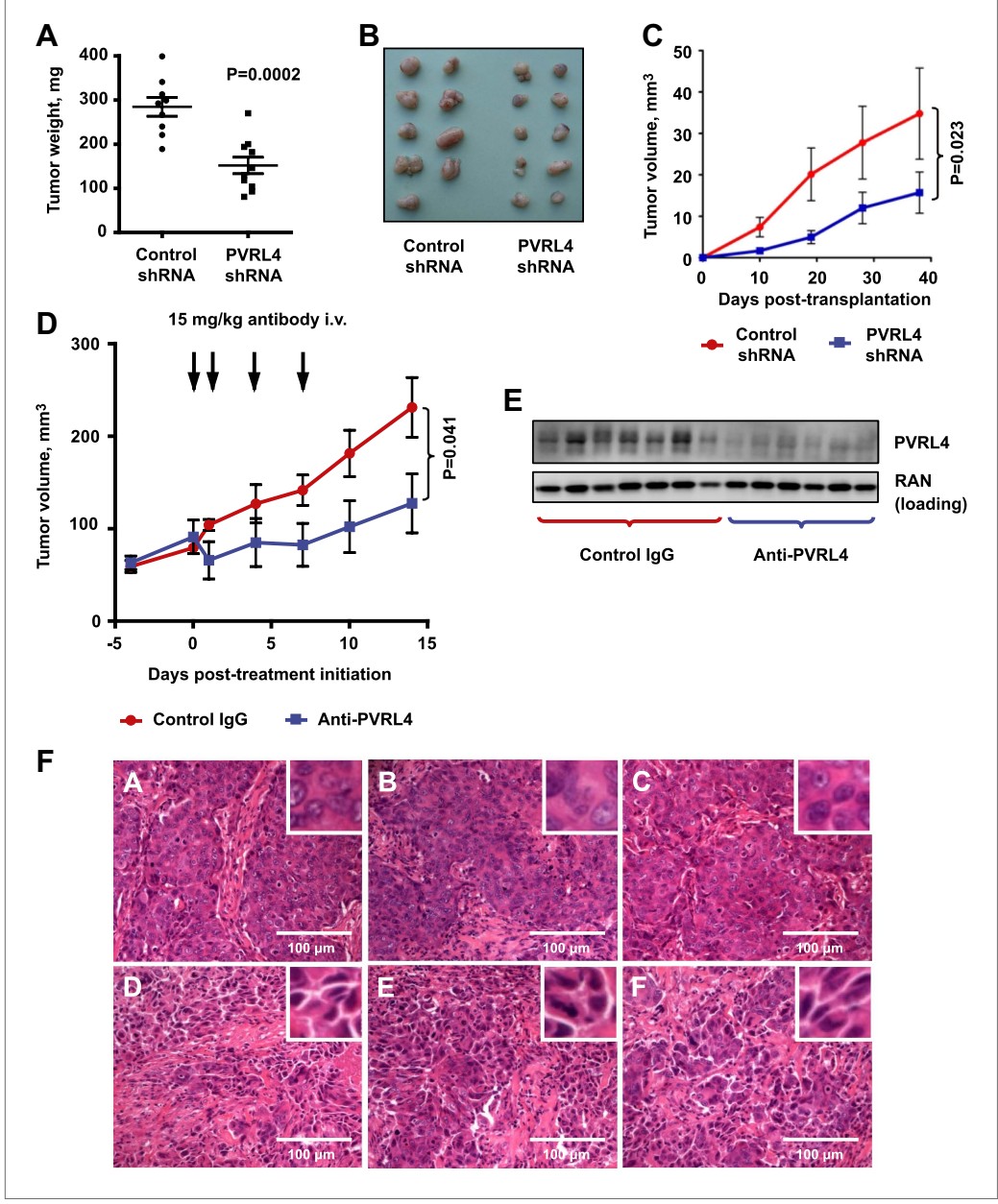

**Figure 7**. Targeting PVRL4 inhibits tumor growth. (**A**) and (**B**) Female nude mice were injected into their mammary fat pads with SUM190 cells expressing PVRL4-targeted or control shRNA (n = 10 per group, error bars ± SEM). The resulting tumors were excised, scaled (**A**), and photographed (**B**). (**C**) SUM185 cells were stably transduced with PVRL4-targeted or control shRNA and injected into the mammary fat pads of female nude mice (n = 10 per group, error bars ± SEM). Tumor volume was measured with calipers at the indicated time points. (**D**) Female nude mice with ~50 mm³ SUM190-eGFP xenografts were randomized into two cohorts (n = 7 per group) and injected with anti-PVRL4 monoclonal antibodies or control IgG on the indicated days. Tumor volume was measured with calipers (error bars ± SEM). (**E**) Levels of PVRL4 protein were measured in tumor lysates from anti-PVRL4 antibody or control-treated mice, 7 days after the last treatment. (**F**) Tumor sections from control IgG (**A**–**C**) or anti-PVRL4 antibody-treated (**D**–**F**) mice were stained with hematoxylin/eosin and photographed. Representative images are shown.

The following figure supplements are available for figure 7:

**Figure supplement 1**. PVRL4 depletion inhibits BT-474 xenograft growth.

*Figure 7. Continued on next page*

*Figure 7. Continued*

**Figure supplement 2**. Anti-PVRL4 antibodies disrupt cell–cell contacts in xenografts in vivo.

**Figure supplement 3**. Anti-PVRL4 antibodies do not induce ADCC in vitro.

**Figure supplement 4**. Anti-PVRL4 antibody treatment does not affect the degree of macrophage infiltration into SUM190 xenografts.

**Figure supplement 5**. Inhibition of PVRL4 by antibodies or by RNAi does not affect expression of EMT markers.

**Figure supplement 6**. Anti-PVRL4 antibodies recognize both human and mouse epitopes.

consecutive intravenous injections of either anti-PVRL4 monoclonal antibody or isotype control IgG at 15 mg/kg each. After the last treatment, we continued to monitor tumor growth for the next 7 days. Whereas tumor volumes in the control group steadily increased over time, the PVRL4 antibody-treated group displayed a remarkable stalling of tumor growth throughout the course of the injection regimen (*Figure 7D*). Immunoblotting of tumor lysates revealed that PVRL4 antibody treatment resulted in a precipitous decline of PVRL4 protein levels, suggesting that cells with the highest expression of PVRL4 were eliminated by the treatment (*Figure 7E*). Dissection of excised xenografts revealed a softer, paste-like composition of PVRL4 antibody-treated tumors compared to the more solid consistency of control-treated samples. This latter observation was corroborated by histological analysis (*Figure 7F*), which revealed areas of widespread necrosis and, importantly, loss of cell contacts (*Figure 7D–F*) when compared to control IgG-treated tumors (*Figure 7A–C*). Finally, examination of three-dimensional tumor architecture in control- and anti-PVRL4 antibody-treated tumors by two-photon confocal microscopy revealed reduced cell–cell contacts in anti-PVRL4 antibody-treated samples as compared to controls (*Figure 7—figure supplement 2*). The loss of cell contacts and necrosis observed in treated tumors supports our model of PVRL4-driven cell-to-cell attachments conferring ability to sustain growth in conditions of matrix anchorage loss.

To test whether an in vivo tumor inhibitory effect of an anti-PVRL4 antibody can be explained by antibody-mediated recruitment of components of innate immunity, we asked whether this antibody was capable of inducing ADCC (antibody-dependent cytotoxicity) in vitro. Specifically, we mixed SUM190 cells with fresh human NK cells and measured the relative degree of cell lysis induced by either anti-PVRL4 antibody or control IgG. No increase in cell lysis was observed with anti-PVRL4 antibody over isotype control-incubated cells, demonstrating that anti-PVRL4 antibody was unable to efficiently recruit Fc receptor-containing cells (*Figure 7—figure supplement 3*). Moreover, immunohistochemical staining of control and anti-PVRL4 IgG-treated tumor sections for a mouse macrophage-specific marker F4/80 revealed no differences in the extent of macrophage infiltration (*Figure 7—figure supplement 4*). Taken together, these data suggest that the observed inhibitory effect is likely to be a consequence of a direct blockade of PVRL4 function.

Taken together, these data suggest that targeting PVRL4-driven tumors with a monoclonal antibody directed towards its extracellular region results in dramatic inhibition of growth and disruption of cell–cell contacts, demonstrating the therapeutic efficacy of such approach. One caveat associated with targeting junctional proteins as an anticancer therapy strategy is that it may inadvertently induce an EMT phenotype and instigate metastasis. To address this, we measured levels of E-cadherin (an epithelial marker) and vimentin (a mesenchymal marker) in SUM190 xenografts treated with anti-PVRL4 antibody in vivo (*Figure 7—figure supplement 5A*) as well as in cultured SUM190 cells in which PVRL4 expression was inhibited by RNAi (*Figure 7—figure supplement 5B*). In both settings E-cadherin and vimentin protein levels remained unchanged, demonstrating that inhibiting PVRL4 in SUM190 cells does not cause EMT; neither does it select for cells with EMT-like characteristics in vivo.

Another safety concern associated with targeting PVRL4 with a monoclonal antibody is that such treatment has the potential to induce damaging effects in tissues normally expressing the antigen. Both human and mouse PVRL4 were strongly recognized by anti-PVRL4 antibody when expressed on the cell surface (*Figure 7—figure supplement 6*). Among normal mouse tissues, PVRL4 expression is

strongest in cornea and skin epidermis (*Wu et al., 2009*). The skin surface of mice treated with anti-PVRL4 antibodies was not visibly affected by treatment, and mice appeared healthy overall, demonstrating that targeted therapy against PVRL4 does not induce acute side effects in this organism. Taken together with our findings that functionally link PVRL4 to tumorigenesis, as well as with its widespread expression in multiple tumor types, targeting PVRL4-driven cell-to-cell attachment can be viewed as a potential therapeutic strategy directed against a broad spectrum of tumor types.

## Discussion

In this study we have performed an unbiased genetic screen to identify genes that facilitate cell growth in the absence of anchorage. The screen identified PVRL4/Nectin-4 as a potent inducer of anchorage-independent growth in a manner that relies upon formation of physical contacts between individual cells. PVRL4 is expressed in a limited set of normal tissues—namely, skin epidermis, hair follicles, placenta, trachea, and lung (*Jelani et al., 2011*)—yet it becomes overexpressed in a large fraction of breast (*Fabre-Lafay et al., 2007*), NSCLC (*Takano et al., 2009*), and ovarian (*Derycke et al., 2010*) tumors. Analysis of a publicly available dataset of copy number alterations involving *PVRL4* in 484 breast tumors indicates a low-level copy number gain in 60.3% of samples, and high-level amplification in another 7.9% of samples (*Cerami et al., 2012*). Moreover, PVRL4 has emerged as a strong predictor of poor postoperative survival of patients with breast and lung cancer (*Takano et al., 2009*; *Athanassiadou et al., 2011*).

We show that PVRL4-driven colony formation is carried out exclusively via the extracellular regions of PVRL4 and its interacting partner PVRL1, while transmembrane and cytoplasmic regions are not required for this phenotype. Cytoplasmic regions of nectins recruit afadin, a large scaffold protein involved in mediating intracellular signaling. Interestingly, and in contrast to PVRL4, loss of afadin has been shown to be strongly associated with poor outcome in breast cancer patients in a number of studies (*Letessier et al., 2007*; *Fournier et al., 2011*). Supporting our findings in TL-HMECs, we found that PVRL4 is required for clonogenic and anchorage-independent growth of breast cancer cells in vitro as well as growth of orthotopically implanted tumors in vivo. In lung cancer cells, siRNA-mediated depletion of PVRL4 was shown to negatively affect adherent growth and motility (*Takano et al., 2009*), highlighting the functional role of PVRL4 across multiple cancer types. Importantly, we were able to reverse the clonogenic defect by expressing a chimeric construct consisting of the extracellular region of PVRL4 fused to the transmembrane domain of CD8, independently verifying that its function is carried out via an extracellular route.

Our findings reveal a potential mechanism of PVRL4-driven anchorage-independent growth. Specifically, we identify integrin β4 as a *cis*-interacting partner of PVRL4 and demonstrate that anchorage-independent growth driven by PVRL4 requires intact integrin β4/SHP-2/SFK signaling. An extensive array of evidence functionally links integrin β4 to breast tumorigenesis. Similarly to PVRL4, integrin β4 expression is associated with poor prognosis and a basal-like expression profile (*Diaz et al., 2005*; *Lu et al., 2008*), and targeting integrin β4 with RNAi was shown to inhibit anchorage-independent growth, invasion, and xenograft growth in breast cancer cell lines (*Lipscomb et al., 2003*; *Bon et al., 2006*). Integrin β4 is thought to exhibit a significant degree of signaling autonomy compared to other integrins, due to its atypically long C-terminus which serves as a scaffold for its downstream effectors. This signaling autonomy was convincingly demonstrated in a study utilizing chimeric constructs where extracellular and transmembrane regions of integrin β4 were substituted with those on the TrkB receptor tyrosine kinase. Dimerization of two such chimeric molecules on the cell surface by adding TrkB ligand, BDNF, was sufficient to drive SFK activation, in a manner dependent on SHP-2 phosphatase (*Merdek et al., 2007*). *Trans*-interactions of cell adhesion molecules are thought to trigger the formation of tightly packed zippers of interacting cell adhesion molecules at a cell–cell contact interface (*Walmod et al., 2004*). One possibility is that *trans*-interacting PVRL4 and PVRL1 similarly organize into zippers at sites of cell–cell contact, driving individual integrin β4 molecules into proximity sufficient for their activation. Matrix attachment-independent activation of integrin molecules may then allow sustained SHP-2/SFK signaling sufficient for counteracting the growth-restrictive consequences of anchorage loss.

In the context of a multicellular tumor, the PVRL4-PVRL1 associated signaling exemplifies a mechanism by which cell-to-cell attachment serves to mimic attachment to matrix, allowing cells to bypass the growth constraint imposed by the requirement for proper anchorage. Interestingly, we found that multiple oncogenic perturbations facilitate increased intercellular adhesiveness. It remains to be further elucidated whether, and through what mechanisms, the increased ability to form contacts functionally contributes to anchorage-independent growth or other tumorigenesis-associated phenotypes

in each of these instances. Increased clustering has been previously associated with greater metastatic capacity (*Updyke and Nicolson, 1986*), raising an interesting possibility that increased cell–cell adhesiveness is a generalized mechanism that tumor cells employ as a part of their survival strategy. It is possible that while loss of cell adhesion during EMT facilitates invasiveness, it concomitantly places cells under the stress of anchorage deprivation, compromising their survival. Thus, cancer cells may need to replace the loss of homotypic cell–cell contacts with weaker heterotypic interactions capable of promoting survival in the absence of anchorage.

In addition to enabling anchorage-independent growth at the primary tumor site, cell–cell clustering in the lymphatic vasculature could promote cell survival. This is particularly relevant for PVRL4 as 100% of breast tumors from patients with two or more lymph nodes that are positive for cancer cells express PVRL4 (*Athanassiadou et al., 2011*). In addition, we demonstrate that PVRL4 can promote the attachment of cancer cells to other cell types. For example, SUM190 breast cancer cells readily attach to lung endothelial cells (HMVEC-Ls), and inhibiting PVRL4 with RNAi or with blocking antibodies abolishes this interaction. CTC-targeted therapy is a promising new treatment possibility, and it will be of particular interest to investigate the potential role of PVRL4 in mediating these types of interactions in vivo.

We directly tested the utility of blocking PVRL4-driven cell–cell contacts with monoclonal antibodies as a potential therapeutic strategy. In vitro, we found that such blocking antibodies suppress PVRL4-driven cellular growth and Src family kinase activation in the absence of anchorage, and in vivo, anti-PVRL4 antibody treatment potently inhibited the growth of orthotopically implanted primary tumors. Importantly, post-antibody therapy tumor samples uniformly exhibit marked downregulation of PVRL4, confirming that the observed response is target-specific and not a consequence of a non-specific anti-tumor effect of the antibody. In addition, we found no difference in the degree of host macrophage recruitment or in vitro ADCC induction between control IgG and anti-PVRL4 antibodies. These observations suggest that the observed tumor growth inhibition is likely due to a direct inhibition of cell-to-cell attachment mediated by PVRL4-PVRL1 binding, and not due to an Fc receptor-mediated immune response. It remains to be tested whether a stronger inhibitory effect could be achieved with an antibody that is both ADCC-competent and capable of inhibiting PVRL4-driven cell clustering.

Even though treatment with anti-PVRL4 antibodies caused dramatic dissolution of cell–cell contacts, we observed no changes in markers of EMT in tumors from mice treated with the antibody; neither did it produce obvious deleterious effects in treated mice. These observations signify not only the potential efficacy but also the safety of the demonstrated approach. Together, these results suggest that anti-PVRL4 monoclonal antibody therapy aimed at disrupting cell–cell interactions may be a viable strategy for treating cancer.

## Materials and methods

### Constructs and virus production

For screen candidate validation, ORFs from isolated colonies were subcloned into pRoles retroviral vector (provided by W Harper). For PVRL4 structure-function analysis, a full-length ORF (accession number BC010423, aa 1-510) or indicated fragments (*Supplementary file 1C*) were generated by PCR and subcloned into pQCXIN retroviral vector (Clontech, Mountain View, CA). PVRL4-CD8 and PVRL1-CD8 chimeric fusions were created the following way: the extracellular region of *PVRL4* (accession number BC010423, aa 1-342) or *PVRL1* (accession number BC113471, aa 1-354) was amplified by PCR using a reverse primer containing a sequence for a 28 aa-long transmembrane domain of the *CD8A* gene (accession number NM_001768, 543–626 bp), followed by a STOP codon. The resulting PCR product was subcloned into the pQCXIN vector. To create an shRNA-resistant PVRL1-CD8 construct, site-directed mutagenesis was performed with the following primers: sh3ResFW 5′-CCAGGCGTCCACAGTCAA GTTGTGCAAGTCAATGACTCCATGTATG-3′ and sh3ResRV 5′-CATACATGGAGTCATTGACTTGCACAAC TTGACTGTGGACGCCTGG-3′ using a QuikChange II Site-Directed Mutagenesis Kit (Agilent, Santa Clara, CA). Stable RNAi-mediated depletion was performed with shRNAs expressed in either pMSCV-PM or pGIPZ vector in a miR-30 context that were either picked from the Hannon-Elledge shRNA library (Open Biosystems, Huntsville, AL) or designed de novo (design and cloning protocol described in *Paddison et al., 2004*). The 21 nt sense sequences of shRNAs used in this study are listed in *Supplementary file 1D*. Tandem HA/FLAG-tagged PVRL4 construct was obtained by Gateway recombination of *PVRL4* entry clone into C-terminal iTAP vector (provided by W Harper). For stable labeling with fluorescent markers,

cells were transduced with pHAGE-dsRed and pMSCV-CMV-GFP viruses. For assaying cell clustering in the presence of oncogenes, TL-HMECs were transduced with pWZL-myr-p110-PI3K-neo or empty pWZL-neo, and pBABE-H-RAS$^{V12}$-puro or empty pBABE-puro. Mouse pSPORT6-PVRL4 was purchased from Open Biosystems. Retro- and lentiviral supernatants were generated by transient transfection of 293T cells following the TransIT transfection protocol (Mirus Bio, Madison, WI) and harvested 48 hr later.

## Cell culture

TL-HMECs expressing hTERT and SV40 Large T antigen (*Zhao et al., 2003*) were cultured in MEGM (Lonza, Allendale, NJ). SUM190 and SUM185 cells (provided by K Polyak) were cultured in a 1:1 mix of MEGM and F12:DMEM (Invitrogen, Carlsbad, CA), supplemented with 5% FBS (Invitrogen). BT-474 and Sk-BR-3 cells were cultured in RPMI-1640 (ATCC, Manassas, VA) with 10% FBS (Invitrogen) and 10 µg/ml of bovine insulin (Sigma, St. Louis, MO). 293T cells were cultured in DMEM (Invitrogen), supplemented with 10% FBS. HMVEC-L cells (Lonza) were cultured in EGM-2 medium (Lonza). Retroviral infections were performed in the presence of 8 µg/ml of polybrene (Sigma). For SUM190 and SUM185 infections, cells were plated in six-well dishes and centrifuged in the presence of viral supernatant and polybrene for 1 hr at 2000 rpm. Successful viral integrants were selected with puromycin (2 µg/ml) or Geneticin (200 µg/ml for TL-HMECs, 750 µg/ml for SUM190).

## Anchorage-independent colony formation and anoikis assays

Anchorage-independent colony formation assays were performed as previously described (*Westbrook et al., 2005*) with minor modifications. Briefly, cells were suspended in reduced growth factor MEGM (containing 50% of kit-supplied BPA, insulin, EGF, and hydrocortisone) with 2% methylcellulose (Sigma) and plated on tissue culture dishes precoated with 0.6% Noble agar (Sigma) in MEM (Invitrogen). For assays performed in 6 cm dishes, $4.5 \times 10^4$ cells per dish were plated. For assays performed in six-well plates, $1.2 \times 10^4$ cells per well were plated. Colonies were counted after 3 weeks of growth. For each assay, an average of three replicates ± SD is shown. For anchorage-independent colony formation assays in the presence of antibodies, the following antibodies were used: normal mouse IgG (MAB004; R&D Systems, Minneapolis, MN) and mouse anti-human PVRL4 IgG2B (MAB2659; R&D Systems) at 4 µg/ml. For anchorage-independent colony formation assays in the presence of PP2 inhibitor, PP2 (EMD Millipore, Billerica, MA) was used at 10 µM final concentration. For filtering of the cell suspension, cells were passed through a nylon mesh 35 µm cell strainer (BD Biosciences, Franklin Lakes, NJ). For anoikis assays, cells were seeded on ultra-low attachment dishes (Corning, Midland, MI) in reduced growth factor MEGM with 1% methylcellulose. For assays performed in 10 cm dishes, $1.4 \times 10^5$ cells were plated; for assays performed in six-well dishes, $2.0 \times 10^4$ cells per well were plated. Total ATP measurements were performed using CellTiter GLO reagent (Promega, Madison, WI) according to the manufacturer's protocol after 72 hr of growth in suspension, and luminescence values were read with a Victor X5 plate reader. For isolation of RNA or protein lysates, cell pellets were harvested after 72 hr of growth in suspension and washed with cold PBS prior to lysis.

## Anchorage-independence screen

A genetic screen for ORFs promoting TL-HMEC anchorage-independent colony formation was performed the following way: TL-HMECs were transduced with a retroviral pool of human open reading frames (ORFeome library V1.1) at a multiplicity of infection of 0.2 and representation of 200. Two independent infections with a complete library were performed. Cells were plated into semi-solid medium (methylcellulose) in the absence of attachment, maintaining the library representation. Resulting colonies were isolated after 3 weeks of growth. A total of 732 colonies were isolated from two screen replicates and individually expanded in 96-well plates. Genomic DNA was isolated from clones and the ORF insert was PCR-amplified and sequenced. A total of 40 candidates (*Supplementary file 1A*) were subsequently subcloned into pRoles and individually tested for their ability to induce colony formation in anchorage-free conditions.

## Clonogenic assays

For assaying clonogenic potential, $1.0 \times 10^3$ SUM190 cells were seeded in 6 cm tissue culture-treated dishes. After 3 wk of growth, the resulting colonies were stained with 1% methylene blue and counted. For each assay, an average of three replicates ± SD is shown.

## Clustering assays

Cells were gently detached off the tissue culture surface with enzyme-free cell dissociation buffer (Invitrogen) and washed once with complete medium. Then $1.0 \times 10^5$ cells were allowed to aggregate in 1 ml of complete medium in a 15 ml conical tube at room temperature. The tubes were gently flicked during the process to visually assess the progression of clustering. After 1–1.5 hr of incubation, the cell suspension was poured into the wells of 12-well dishes and allowed to attach to the bottom of the dish for 5–10 min. Cells were promptly visualized and photographed under phase contrast and fluorescent filters using an AxioVert inverted microscope. When clustering assays were performed in the presence of antibodies, the following antibodies were used: normal mouse IgG (MAB004; R&D Systems), normal goat IgG (AB-108-1C; R&D Systems), mouse anti-human PVRL4 IgG2A (MAB26591; R&D Systems), mouse anti-human PVRL4 IgG2B (MAB2659; R&D Systems), goat anti-human PVRL4 (AF2659; R&D Systems), goat anti-human PVRL1 (AF2880; R&D Systems), and DECMA-1 (anti-human E-cadherin, ab11512; Abcam, Eugene, OR). All antibodies were used at a concentration of 4 µg/ml. For short-term culture of TL-HMECs in suspension, $4.0 \times 10^5$ cells were allowed to aggregate in 1 ml of complete medium in a 15 ml conical tube at room temperature, mixed with 5 ml of 0.5% methylcellulose in reduced growth factor MEGM and incubated in the wells of a six-well ultra-low attachment dish (two wells per sample) for indicated periods of time.

## Western blotting

Cells were lysed in NP-40 buffer (1% NP-40, 25 mM Tris–HCl, pH 7.4, 150 mM NaCl, 1 mM EDTA, 10% glycerol) in the presence of protease and phosphatase inhibitor tablets (Roche, Indianapolis, IN). Adherent cells were lysed for 15 min on ice, followed by scraping into Eppendorf tubes and centrifugation at 14,000 rpm for 15 min at 4°C. Suspension-cultured cells were washed once with cold PBS, followed by lysis and centrifugation. The protein concentration in supernatants was measured using the BCA assay (Pierce, Rockford, IL) and lysates were brought to identical concentrations with lysis buffer. Samples were mixed 1:1 with 2× Laemmli buffer (125 mM Tris–HCl, pH 6.8, 4% SDS, 20% glycerol, 0.004% bromophenol blue) and DTT was added to final concentration of 25 mM. Samples were boiled for 8 min and loaded on Tris-glycine 4–20% or 4–12% gradient gels (Invitrogen). Transfer/blotting was performed as described elsewhere. Western blotting was performed with the following antibodies: goat anti-PVRL4 (AF2659; R&D Systems), goat anti-PVRL1 (AF2880; R&D Systems), mouse anti-Vinculin (V9131; Sigma), mouse anti-RAN (610340; BD Biosciences), mouse anti-ITGB4 (611232; BD Biosciences), mouse anti-FLAG-HRP (A8592; Sigma-Aldrich, St. Louis, MO), rabbit anti-pY416 SFK (2101; Cell Signaling, Danvers, MA), mouse anti-SFK (2110; Cell Signaling), mouse anti-SHP-2 (610621; BD Biosciences), and mouse anti-α Tubulin (sc-8035; Santa Cruz, Santa Cruz, CA).

## HA pulldown

TL-HMECs expressing HA/FLAG-PVRL4 or HA/FLAG-GFP were gently detached off the tissue culture surface with enzyme-free cell dissociation buffer, washed, and lysed for 30 min in MCLB lysis buffer (50 mM Tris, pH 7.5, 150 mM NaCl, 1% NP-40) in the presence of protease and phosphatase inhibitors (Roche), followed by centrifugation at 14,000 rpm at 4°C. Lysates were precleared with protein A/G beads (sc-2003; Santa Cruz). Pulldowns were performed with anti-HA beads (A2095; Sigma-Aldrich) overnight at 4°C. Beads were washed five times with MCLB buffer, followed by two washes with elution buffer (50 mM Tris, pH 7.5, 150 mM NaCl, 10% glycerol). Elutions were performed with 500 µg/ml of HA peptide (I2149; Sigma-Aldrich) in elution buffer. Proteins were precipitated from this mixture with 20% trichloroacetic acid (TCA), and the resulting pellet was washed once with 10% TCA and four times with cold acetone.

## Mass spectrometry

TCA-precipitated proteins were dissolved in 100 mM ammonium bicarbonate (pH 8.0) with 10% acetonitrile and 10 ng/µl trypsin (Promega) and incubated at 37°C for 5 hr. They were subsequently desalted, dissolved in 5% formic acid/5% acetonitrile, and loaded onto a reversed phase microcapillary column (100 mm I.D.) packed with 18 cm of Maccel C18AQ resin (3 mm, 200 A; The Nest Group, Southborough, MA). Peptides were eluted using a gradient of 4–26% acetonitrile in 0.125% formic acid over 95 min and detected in a hybrid linear ion trap-orbitrap mass spectrometer (LTQ-Orbitrap Discovery; ThermoFisher, West Palm Beach, FL). Precursors selected for MS/MS fragmentation were

corrected for errors in monoisotopic peak assignment, and tandem MS spectra were searched using the Sequest algorithm, with mass tolerance set to 25 ppm and two missed cleavages allowed. False discovery rates were estimated with the target-decoy method (*Elias and Gygi, 2007*), and linear discriminant analysis (LDA) was utilized to filter peptides to an initial 1% peptide-level false discovery rate (FDR). Peptides were then assembled into proteins and further filtered to a protein-level FDR of 0.84% (*Huttlin et al., 2010*), resulting in a final peptide-level FDR of 0.35%.

## Immunoprecipitations

Cells were lysed in IP lysis buffer (20 mM Tris–HCl, pH 8.0, 150 mM NaCl, 1 mM $CaCl_2$, 1 mM $MgCl_2$, 10% glycerol, 1% NP-40) in the presence of protease and phosphatase inhibitors and centrifuged at 14,000 rpm at 4°C. For the immunoprecipitation of integrin β4, cells were pretreated with a reversible, cell-impermeable homobifunctional cross-linking agent DTSSP (21,578; Pierce) at 1 mM concentration for 45 min on ice. The cross-linking agent was then quenched with 20 mM glycine for 15 min and cells were washed twice with PBS, followed by lysis in IP lysis buffer. Immunoprecipitations were performed with anti-HA beads (A2095; Sigma-Aldrich) for 2 hr or with anti-ITGB4 antibody (555722; BD Biosciences) for 2 hr, followed by protein A/G beads (sc-2003; Santa Cruz) for 1 hr. Beads were washed four times with lysis buffer and boiled in Laemmli buffer with 25 mM DTT.

## RT-qPCR

Total RNA was isolated from cells using the RNAeasy Plus kit (QIAgen, Germantown, MD). cDNA was synthesized from 1 μg of total RNA, using Superscript III Reverse Transcriptase (Invitrogen) and Oligo-dT primer (Invitrogen), following the manufacturer's protocol. Quantitative PCR was performed with the Platinum SYBR Green qPCR Supermix-UDG kit (Invitrogen) on an Applied Biosystems 7500 Fast Real Time PCR machine. Gene-specific primers were designed using the Universal Probe Library (Roche Applied Science, Penzberg, Germany). PCR reactions were carried out in triplicates or quadruplicates. For each value, an average of at least three replicates ± SD is shown. Sequences of gene-specific primers used for qPCR are listed in *Supplementary file 1E*.

## Immunofluorescence

TL-HMEC cells were cultured on chamber slides (BD Biosciences) and fixed with cold methanol following by blocking in blocking buffer containing BSA and cold water fish gelatin (Sigma-Aldrich). Cells were incubated with primary goat anti-PVRL4 antibody (AF2659; R&D Systems), at 1:500 dilution at 4°C overnight, followed by 1-hr incubation with chicken anti-goat Alexa-Fluor 488 secondary antibody (Invitrogen) at 1:2500 dilution. Cells were mounted in Vectashield mounting medium with DAPI (Vector Labs, Burlingame, CA) and visualized under a fluorescent microscope.

## Live-cell FACS analysis

293T cells were gently detached off the adherent surface with enzyme-free cell dissociation buffer (Invitrogen) and washed once with serum-free DMEM. Cells were incubated with 10% normal goat serum (Invitrogen) for 10 min at room temperature, then with a primary antibody at 1:100 dilution in 5% goat serum for 30 min on ice, followed by a goat anti-mouse secondary antibody coupled to Alexa Fluor 488 (Invitrogen) at 1:1000 dilution in PBS for 30 min on ice. The fluorescent signal was measured on an LSR II FACS Analyzer and analyzed with FlowJo software.

## In vivo experiments

All mouse experiments were performed with the approval of the Massachusetts Institute of Technology (MIT) Committee on Animal Care. For subcutaneous xenograft assays with shRNA-transduced cells, nude mice (Taconic) were injected orthotopically into the mammary fat pad with $1.0 \times 10^6$ SUM190 or SUM185 cells, or subcutaneously with $1.0 \times 10^7$ BT-474 cells in serum-free medium with 50% Matrigel (BD Biosciences), one injection site per mouse. For the BT-474 xenograft experiment, mice were implanted with a 60 d release pellet containing 0.72 mg of 17β-estradiol (Innovative Research of America, Sarasota, FL) 72 hr prior to cell injections. Tumor growth was monitored by caliper measurements. Tumors were excised after 4 wk of growth, scaled, and photographed. For the antibody treatment experiment, 15 mg/kg of anti-PVRL4 antibody or normal IgG control (R&D Systems) was injected 4 times at days 0, 1, 4, and 7. Treatment response was assessed by caliper measurements of the tumors in anesthetized mice. Mice were euthanized 15 d after treatment initiation and tumors were harvested and representative parts subjected to paraffin

embedding, Western blot analysis, and direct microscopy on an inverted Olympus Multiphoton Laser Scanning Confocal Microscope using a 25× objective. Images were analyzed by the IMARIS software package.

## ADCC assay

PVRL4 antibody-dependent cell-mediated cytotoxicity was assessed by using europium-labeled (DELFIA cell cytotoxicity assay; PerkinElmer, Waltham, MA) SUM190 cells as target cells in a 96-well format. Primary human NK cells were isolated from a fresh buffy coat using magnetic purification (EasySep Human CD56 Positive Selection Kit; STEMCELL Technologies). Cells were co-incubated at a 1:1, 1:10, and 10:1 effector-to-target cell ratio. Next, 25 µg/ml anti-PVRL4 antibody was added to the respective wells. Cells were co-incubated for 4 hr and cell-free supernatants further subjected to time-lapsed fluorescence quantification using a TECAN Infinite 200 PRO plate reader. To determine the maximal range of europium release, target cells were lysed in DELFIA lysis buffer. Levels of released europium are displayed as percentage values of maximum release determined by lysis. To control for NK cell effector functionality, the ADCC assay was performed using the hMB lymphoma model (*Leskov et al., 2012*) as a target cell line in the presence of 25 µg/ml of anti-CD52 antibody (alemtuzumab).

## Immunohistochemistry

Immunohistochemical staining with an anti-mouse F4/80 antibody (MCA497GA; Serotec, Raleigh, NC) was performed using an Envision kit (Dako, Glostrup, Denmark). Retrieval was performed by incubating slides in Proteinase K (Dako) at a 1:5 dilution for 10 min. The primary antibody was used at a 1:10,000 dilution for 60 min at room temperature.

## Acknowledgements

We thank the Specialized Histopathology Core-Longwood for performing the immunohistochemistry staining. We thank Dr W Harper for the pRoles vector, Dr N Polyak for SUM190 and SUM185 cells, and N Polyak, J Brugge, and C Kim as well as members of the Elledge Lab for helpful discussions and critical reading of this manuscript.

## Additional information

### Funding

| Funder | Author |
| --- | --- |
| Department of Defense BCRP | Stephen Elledge |
| Howard Hughes Medical Institute | Stephen Elledge |

The funders had no role in study design, data collection and interpretation, or the decision to submit the work for publication.

### Author contributions

NNP, Conception and design, Acquisition of data, Analysis and interpretation of data, Drafting or revising the article; CP, Performed animal experiments and data analysis/interpretation, Acquisition of data, Analysis and interpretation of data, Drafting or revising the article; AEHE, Performed mass spectrometry analysis, Acquisition of data, Analysis and interpretation of data, Drafting or revising the article; CJB, Acquisition of data, Analysis and interpretation of data; TFW, Designed and helped perform the screen for anchorage-independence, Conception and design, Acquisition of data, Analysis and interpretation of data, Drafting or revising the article; MH, Conception and design, Drafting or revising the article; SJE, Conception and design, Analysis and interpretation of data, Drafting or revising the article

### Ethics

Animal experimentation: Animals used in this study were housed in designated facilities that are registered with the US Department of Agriculture, and the animal program as a whole is fully accredited by AAALACi. Massachusetts Institute of Technology (MIT) has an Animal Welfare Assurance on file with the Office for Protection from Research Risks. Oversight of the health and well-being of animals maintained at MIT is the primary responsibility of the clinical veterinary staff of MIT's Division of

Comparative Medicine (DCM). All animal usage in this study was conducted according to protocol 0512-041-15 approved by MIT's Committee on Animal Care (CAC), whose Assurance Number (IACUC) is A-31125-01. The CAC inspects animals, animal facilities, and laboratories, and reviews all research and teaching exercises which involve vertebrate animals or other tissues before activities are performed. Animal protocols are renewed every three years and reviewed annually. Besides the staff and certified veterinarians provided by MIT's DCM, all investigators working with animals are trained by DCM for proper animal welfare. Animals are inspected daily for signs of infection and distress.

## Additional files

### Supplementary files

• Supplementary file 1. (**A**) List of candidate ORFs from the anchorage-independence screen. (**B**) List of validated ORFs which localize to statistically defined peaks of focal amplification in human tumors. (**C**) PVRL4 deletion series. (**D**) Sequences of shRNAs used in the study. (**E**) Sequences of qPCR primers used in the study.

### Major datasets

The following previously published datasets were used:

| Author(s) | Year | Dataset title | Dataset ID and/or URL | Database, license, and accessibility information |
|---|---|---|---|---|
| Beroukhim R, Mermel CH, Porter D, Wei G, Raychaudhuri S, Donovan J, et al. | 2010 | Data from: The landscape of somatic copy-number alteration across human cancers. Nature 463, 899-905. doi: 10.1038/nature08822 | http://www.broadinstitute.org/tumorscape/pages/portalHome.jsf | Publicly available at Tumorscape (www.broadinstitute.org/tumorscape/) |
| Cancer Genome Atlas Network | 2012 | Comprehensive molecular portraits of human breast tumors | http://www.cbioportal.org/public-portal/study.do?cancer_study_id=brca_tcga_pub and https://tcga-data.nci.nih.gov/docs/publications/brca_2012/ | Publicly available at cBioPortal for Cancer Genomics (http://www.cbioportal.org/) and The Cancer Genome Atlas Data Portal (https://tcga-data.nci.nih.gov/tcga/) |

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
