## [Decision Letter]

Thank you for choosing to send your work entitled “A Role for PVRL4-driven Cell–Cell Interactions in Tumorigenesis” for consideration at *eLife*. Your article has been evaluated by a Senior editor and 2 reviewers, one of whom is a member of our Board of Reviewing Editors.

The Reviewing editor and the other reviewer discussed their comments before we reached this decision, and the Reviewing editor has assembled the following comments based on the reviewers' reports to help you prepare a revised submission.

Pavlova et al present the result of an ORFeome screen for genes that confer anchorage-independent growth on TL-HMEC cells, a model of anchorage-dependent breast cancer. A cDNA for PVRL4/Nectin-4 was repeatedly recovered from this screen. PVRL4 not only caused anchorage-independent growth but also caused strong aggregation of the TL-HMEC cells. They note copy number gains of the genomic region encompassing PVRL4 in some cases of human breast cancer (∼8%). Knockdown of PVRL4 decreased clonogenic survival of one breast cancer line with PVRL4 amplification in the absence of stroma and delayed tumor progression in xenografts. An antibody to PVRL4 decreased xenograft progression, apparently by preventing cell-cell adhesive contacts.

This study is notable for the elegant ORFeome screen to investigate anchorage-independent growth. PVRL4 appears to be capable of mediating anchorage-independent growth and antibodies to PVRL4 would seem to be a viable therapeutic strategy in a cancer with very high levels of PVRL4 expression. However, more analysis is needed to determine how often human breast cancers (or other cancer types) utilize the PVRL4 pathway oncogenically.

Major points:

1) It is unclear how often PVRL1 is involved in focal and high-level amplifications. The authors state that: “Analysis of a publicly available dataset of copy number alterations from 541 breast tumors identified high-level amplification of PVRL4 locus in 6.5% of samples (Cerami et al., 2012).” This website only lists samples as having gain, loss, or wild-type loci, but does not distinguish high-level amplification from single copy gain. In the Oncomine database, arrayCGH data from 650 invasive ductal breast cancers is available along with a measure of copy number. The PVRL4 copy number alterations appear to be single or low copy number gains rather than high-level amplifications. For example, in this set of tumors, HER2 and CCND1 were amplified > 4N in 9% and 3% of cases, but there are no cases that had this level of amplification of PVRL4. Moreover, while a few cases have focal gains involving PVRL4, the vast majority of gains involving this locus involve broad chromosomal regions. Thus, the evidence from copy number does not strongly support PVRL1 as a driver oncogene in most breast cancers.

2) Of note, however, many breast cancers seem to over express PVRL4 compared to normal breast epithelium. While overexpression is not as strong evidence of pathogenetic importance, it can be an important oncogenic mechanism in some instances. In this regard, a study of PVRL4 levels by FACS (Fabre-Lafay et al. BMC Cancer 2007) showed particularly high levels in the two cell lines that the authors studied, namely SUM190 (m.f.i. 500) and SUM185 (m.f.i. 145). However, the majority of breast cancer lines had much lower levels. For example, the commonly studied breast cancer line MCF-7 had a PVRL4 m.f.i of 22. It would therefore be informative to investigate the role of PVRL4 in a cell line with this lower level of expression, both in terms of in vitro anchorage-independent growth and using xenograft to determine the effect of PVRL4 knockdown and anti-PVRL4 antibodies.

---

## [Author Response]

*1) It is unclear how often PVRL1 is involved in focal and high-level amplifications. The authors state that: “Analysis of a publicly available dataset of copy number alterations from 541 breast tumors identified high-level amplification of PVRL4 locus in 6.5% of samples (Cerami et al., 2012).” This website only lists samples as having gain, loss, or wild-type loci, but does not distinguish high-level amplification from single copy gain. In the Oncomine database, arrayCGH data from 650 invasive ductal breast cancers is available along with a measure of copy number. The PVRL4 copy number alterations appear to be single or low copy number gains rather than high-level amplifications. For example, in this set of tumors, HER2 and CCND1 were amplified > 4N in 9% and 3% of cases, but there are no cases that had this level of amplification of PVRL4. Moreover, while a few cases have focal gains involving PVRL4, the vast majority of gains involving this locus involve broad chromosomal regions. Thus, the evidence from copy number does not strongly support PVRL1 as a driver oncogene in most breast cancers*.

We appreciate the reviewers’ insightful criticism with regard to the nature of genomic alterations to PVRL4 in human tumors. As the reviewers point out, in the 650-tumor dataset from the Oncomine database, arrayCGH data indicate that PVRL4 in tumors is affected mostly by low-level copy gains, but not by high-level amplifications that are seen for HER2 and CCND1. In agreement with the Oncomine dataset, another cancer genomic alterations database, cBioportal, which we cite in our manuscript, similarly indicates that PVRL4 is less often affected by high-level amplifications than HER2. However, we do not completely agree with the reviewers’ statement that cBioportal “does not distinguish high-level amplification from single copy gain”. Indeed, in cBioportal, an arbitrary score is assigned to indicate the amplitude of a genomic alteration that involves the gene of interest, with “0” indicating a diploid status, “1” a low-level copy gain, and “2” a high-level amplification.

According to this scoring system, in the case of PVRL4, 60.3% of breast tumors contain low-level copy gains and 7.9% contain high-level amplifications, whereas in case of HER2, 13.9% of breast tumors have low-level copy gains, and 13.2% have high-level amplifications. To address the reviewers’ concerns, we have changed the wording we use to describe the genomic status of PVRL4 and expanded our statement on the status of PVRL4 in tumors to reflect the data more accurately: “Analysis of a publicly available dataset of copy number alterations involving PVRL4 in 484 breast tumors indicates a low-level copy number gain in 60.3% of samples, and high-level amplification in another 7.9% of samples”.

We would also like to point out that while PVRL4 is not quite as amplified as HER2, it is in the ballpark and the degree of amplification of any particular oncogene depends on many factors, such as genomic structure at the locus and other mechanisms through which overexpression can be achieved. For example, transcription upregulation or protein stabilization could also play important roles in driving PVRL4 expression in 60% of breast cancers, in addition to copy number elevation.

We fully agree with the reviewers’ statement that neither genomic nor protein expression evidence, in isolation, constitute proof of PVRL4 being a driver oncogene. However, we feel that published genomic and protein expression data from human cancer can be viewed as lending support to the substantive body of functional evidence that we present in this work, and altogether serve to implicate PVRL4 in the promotion of tumorigenesis.

*2) Of note, however, many breast cancers seem to over express PVRL4 compared to normal breast epithelium. While overexpression is not as strong evidence of pathogenetic importance, it can be an important oncogenic mechanism in some instances. In this regard, a study of PVRL4 levels by FACS (Fabre-Lafay et al. BMC Cancer 2007) showed particularly high levels in the two cell lines that the authors studied, namely SUM190 (m.f.i. 500) and SUM185 (m.f.i. 145). However, the majority of breast cancer lines had much lower levels. For example, the commonly studied breast cancer line MCF-7 had a PVRL4 m.f.i of 22. It would therefore be informative to investigate the role of PVRL4 in a cell line with this lower level of expression, both in terms of in vitro anchorage-independent growth and using xenograft to determine the effect of PVRL4 knockdown and anti-PVRL4 antibodies*.

We thank the reviewers for suggesting that we test the pro-tumorigenic role of PVRL4 in a panel of breast cancer cell lines, in which not only those cell lines with high PVRL4 expression (such as SUM190 and SUM185 cells), but also cell lines in which PVRL4 is expressed at the medium level, would be represented. To address this, we tested the effect of PVRL4 depletion using two independent shRNAs in two extra breast cancer cell lines, Sk-BR-3 and BT-474 (m.f.i. of 75 and 40, respectively – thus falling into a medium range of protein expression, according to FACS analysis performed by Fabre-Lafay et al., 2007).

We found that both Sk-BR-3 and BT-474 displayed defects in clonogenic growth and anchorage-independence phenotype upon PVRL4 depletion. We opted not to test the effect of an anti-PVRL4 antibody treatment on BT-474 xenografts due to the costs associated with performing an additional antibody experiment in mice, and focused on testing effects of RNAi-mediated depletion of PVRL4 in this line instead, as suggested. Indeed, we found that xenograft growth of BT-474 cells was markedly sensitive to PVRL4 depletion, which was similar to what we have observed with SUM190 and SUM185 cell lines. Altogether, our data indicate that tumors with high PVRL4 and those with intermediate expression of PVRL4 may be sensitive to anti-PVRL4 therapy.